# Anticancer Potential of Natural Chalcones: In Vitro and In Vivo Evidence

**DOI:** 10.3390/ijms241210354

**Published:** 2023-06-19

**Authors:** Radka Michalkova, Ladislav Mirossay, Martin Kello, Gabriela Mojzisova, Janette Baloghova, Anna Podracka, Jan Mojzis

**Affiliations:** 1Department of Pharmacology, Faculty of Medicine, Pavol Jozef Šafárik University, 040 01 Košice, Slovakia; radka.michalkova@upjs.sk (R.M.); martin.kello@upjs.sk (M.K.); 2Center of Clinical and Preclinical Research MEDIPARK, Faculty of Medicine, Pavol Jozef Šafárik University, 040 01 Košice, Slovakia; gabriela.mojzisova@upjs.sk; 3Department of Dermatovenerology, Faculty of Medicine, Pavol Jozef Šafárik University, 040 01 Košice, Slovakia; janette.baloghova@upjs.sk (J.B.); anna.podracka@upjs.sk (A.P.)

**Keywords:** natural chalcones, antiproliferative, anticancer, solid cancers

## Abstract

There is no doubt that significant progress has been made in tumor therapy in the past decades. However, the discovery of new molecules with potential antitumor properties still remains one of the most significant challenges in the field of anticancer therapy. Nature, especially plants, is a rich source of phytochemicals with pleiotropic biological activities. Among a plethora of phytochemicals, chalcones, the bioprecursors of flavonoid and isoflavonoids synthesis in higher plants, have attracted attention due to the broad spectrum of biological activities with potential clinical applications. Regarding the antiproliferative and anticancer effects of chalcones, multiple mechanisms of action including cell cycle arrest, induction of different forms of cell death and modulation of various signaling pathways have been documented. This review summarizes current knowledge related to mechanisms of antiproliferative and anticancer effects of natural chalcones in different types of malignancies including breast cancers, cancers of the gastrointestinal tract, lung cancers, renal and bladder cancers, and melanoma.

## 1. Introduction

Cancer is one of the leading causes of death worldwide with 10 million deaths in 2020 [1]. Moreover, according to GLOBOCAN 2020, a 47% increase in the incidence of cancer disease is expected over the next 20 years [2]. Despite significant progress in the diagnosis and treatment of cancer diseases, cancer still remains a disease that is often difficult to cure [3]. In addition, “classical” chemotherapy is often associated with severe toxicity, mostly resulting from low selectivity against cancer cells. Therefore, the discovery of new, effective compounds with low toxicity still remains a great challenge.

Plants are an amazing source of active compounds with a significant impact on human health. The effects of plant products on health are primarily determined by the presence of secondary metabolites produced by the plant for protection against various external biotic and abiotic factors. Among them, polyphenols have attracted the attention of researchers due to their diversity of biological effects [4]. Based on their chemical structure, polyphenols are categorized into two classes: flavonoids and non-flavonoids. Of these, flavonoids are the most prevalent polyphenols, with several thousand members [5]. They can be divided into different groups, such as flavones, flavonols, flavanones, isoflavones, and anthocyanidins. In addition to these basic flavonoids, there are other minor groups, including chalcones and dihydrochalcones [6].

Chalcones, 1,3-diphenyl-2-propene-1-ones, are precursors for flavonoids and isoflavonoid synthesis in plants. They have relatively simple chemical structures but exhibit a great diversity of actions. The chemical structures of some chalcones cited in the text are shown in Figure 1.

Except for plant-related functions, chalcones have been found to possess a broad spectrum of biological activities also in humans including antimicrobial, antidiabetic, cardioprotective, anti-inflammatory, neuroprotective, and antioxidant [7,8,9] effects (Figure 2). In addition to these actions, chalcones have also attracted attention due to their ability to modulate the process of carcinogenesis at different levels [10,11]. Regarding anticancer effect, we and others have reported the growth-inhibitory effect of numerous chalcones in vitro [12,13,14,15] or the anticancer effect in vivo [16,17]. Additionally, multimodal pharmacological effects of chalcones on cellular and molecular levels have been recently reviewed [18]. In the present paper, we discuss the antiproliferative and anticancer effects of natural chalcones in solid cancers, including their cellular and molecular mechanisms of action.

## 2. Chalcones and Breast Cancer

Breast cancer is a malignancy preferentially affecting women with high incidence and mortality rates. It is a complex, highly heterogeneous disease with two kinds of tumor heterogeneity: inter-tumor and intra-tumor heterogeneity [19]. Therefore, classification of breast cancer is complicated and includes primary tumor, lymph node and metastasis evaluation as well as important molecular markers. These biological factors imply immunohistochemical markers, e.g., estrogen and progesterone receptor (ER, PR), human epidermal growth factor receptor 2 (HER2), and proliferation marker protein Ki-67; genomic markers, e.g., breast cancer susceptibility proteins (BRCA1, BRCA2), and phosphatidylinositol-4,5-bisphosphate 3-kinase catalytic subunit alpha (PIK3CA); and immunomarkers, e.g., tumor-infiltrating lymphocytes (TILs) and programmed death-ligand 1 (PD-L1) and receptors [20]. However, among others, triple-negative breast cancer (TNBC) and inflammatory breast cancer (IBC) are considered the most aggressive types of breast cancer, with no significant treatment improvement [21]. The major obstacle to achieving effective treatment is resistance to therapies. Therefore, the therapy options and search for novel effective agents remain current. Chalcones have been shown to express different activities suppressing malignant cells, including breast cancer cell proliferation, adhesion, migration and ability to metastasize. This is why they have been classified as promising anticancer agents in breast cancer.

### 2.1. Xanthohumol

Xanthohumol (XH) is a prenylated chalcone known for being isolated from the hops plant (*Humulus lupulus*) and for its numerous biological activities, such as antimicrobial, antiviral, anti-inflammatory, immunomodulatory, and anticancer effects [22,23].

Besides other human tumors, it inhibits the proliferation of different breast cancer cells. The effect of XH has been investigated in both in vitro and in vivo studies. Initial studies in MCF7 cell lines revealed that XH decreased cell proliferation in a dose-dependent manner. The following oral administration to nude mice inoculated with MCF7 cells resulted in central necrosis within tumors, reduced inflammatory cell number and increased percentage of apoptotic cells. Anti-angiogenic effect of XH was approved by decreased microvessel density and factor VIII expression in XH-treated tumors. Decreased NF-κB activity, phosphorylated inhibitor of kappa B, and interleukin-1b also indicated that XH was able to target both breast cancer and inflammatory cells, as well as endothelial cells [24]. In addition, the actions of XH should be also mediated by the effect on cellular mitochondrial function. This was demonstrated by measuring MCF7 cell viability and reactive oxygen species (ROS) production. XH exerted a dual effect on these breast cancer cells. In low doses (0.01 mM) XH decreased ROS production along with an increase in oxidative phosphorylation system expression. In contrast, high concentrations of XH (5 mM) resulted in increased ROS production, which was accompanied by a decrease in oxidative phosphorylation system expression. The authors suggested that XH in a low dose probably improved, whereas a high dose worsened the mitochondrial function [25]. Mitochondrial effects of XH were then evaluated in another study with the MDA-MB-231 breast cancer cell line. Treatment of cells with 10 μM and 20 μM XH for 48 h significantly decreased the rate of survival with a concurrent increase in the number of sub G0/G1 cells. Observed DNA fragmentation was accompanied by increased expression of Bax and the activity of caspase-3 and -9 in the mitochondria, suggesting the inhibition of cell proliferation through mitochondria- and caspase-dependent apoptotic pathways [26].

In ERα-positive breast cancer cells, XH also works by inhibiting tumor suppressor protein prohibitin 2 binding to brefeldin A inhibited guanine nucleotide-exchange protein 3 (BIG3). This inhibition released prohibitin 2 to directly bind to both nuclear and cytoplasmic ERα. The effect was followed by complete suppression of the estrogen-signaling pathways and ERα-positive breast cancer cell growth both in vitro and in vivo. No such effects were observed in normal mammary epithelial cells [27]. Besides ERα-positive breast cancer cells, XH also effectively inhibited the growth and proliferation of TNBC cells. Evaluation of different chalcones have shown that treatment with chalcone, 2-hydroxychalcone, and XH for 24 h inhibited the growth of MDA-MB-231 cells with IC_50_ values of 18.1, 4.6, and 6.7 μM, respectively. Similar cytotoxicity was observed in another TNBC cell line, Hs578T. Both XH and 2-hydroxychalcone induced apoptosis by Bcl-2 downregulation. Moreover, they exerted more potent inhibitory effects on the proliferation, matrix metalloproteinase (MMP-9) expression and invasive phenotype of MDA-MB-231 than chalcone [28]. Similar results were obtained by comparing MCF-7 and MDA-MB-231 cells in vitro and in 4T1 breast tumor mouse in vivo model. XH significantly decreased cell viability, induced G0/G1 cell cycle arrest and apoptosis in vitro and inactivated the Notch signaling pathway, Notch 1 and Ki-67, in vivo [29]. In vivo studies on a BALB/c-4T1 breast cancer mouse model confirmed antiproliferative effects of XH in another TNBC cell line. In addition to decreased proliferation, XH inhibited expression of proliferation marker protein Ki-67 as well as breast cancer-specific marker cancer antigen 15-3 in this highly tumorigenic and invasive cell line. At the same time, XH enhanced the secretion of perforin, granzyme B, increased the ratio of CD8+/CD25+ and upregulated Th1 cytokines, while it downregulated Th2 cytokines, resulting in a markedly elevated ratio. Because Th1/Th2 is closely related to a variety of tumors, XH-induced enhancement of Th1 immunity response could appear as a more effective approach in mediating anticancer function [30].

Development of drug resistance, which usually goes together with resistance to ionizing radiation results in serious breast cancer treatment problems. To find a method to overcome this obstacle, several natural and synthetic compounds, including XH, were investigated. When the effect of XH on doxorubicin-resistant breast cancer cells (MCF-7/ADR) was investigated, it inhibited viability, induced apoptosis, and arrested the cell cycle of these cells in a dose-dependent manner. Moreover, it increased the inhibitory effect of doxorubicin on MCF-7/ADR cells. These processes were accompanied by decreased colony formation, migration, sphere formation, and the downregulation of stemness-related biomarkers. All these effects indicated the efficiency of XH in the regulation of stemness of doxorubicin-resistant breast cancer cells [31]. The main aim of the experiment with XH, which was performed on MCF-7 and MCF-7/ADR cells, was to determine its radio-sensitizing activity. The results revealed that pretreatment of the cells with XH significantly sensitized MCF-7/ADR cells to radiation treatment by inducing apoptosis. The effect was accompanied by decreased expression of anti-apoptotic proteins, such as MDR1, epidermal growth factor receptor (EGFR) and signal transducer and activator of transcription 3 (STAT3) and increased expression of death receptors DR4 and DR5. All these results suggest that XH pretreatment restores the sensitivity of MCF-7/ADR cells to doxorubicin and radiation therapies [32].

### 2.2. Butein

Butein, 2′,3,4,4′-tetrahydroxychalcone, found in several plants such as the heartwood of *Dalbergia odoriferau* or the stem bark of *Semecarpus anacardium*, has been reported to possess several beneficial properties, such as antioxidant, anti-inflammatory, and antimicrobial activities [33,34]. Moreover, numerous studies have revealed that butein exhibits antiproliferative and anticancer effects in vitro and in vivo [35].

Direct actions of butein on breast cancer cell proliferation result from modulation of the effect of different cellular proteins. It has been suggested that cyclooxygenase (COX) inhibitors could be used in the chemoprevention of breast carcinogenesis. In addition, COX-2 isoform was found to be overexpressed in breast cancer tissues. Butein (at or below 10 μM) downregulated phorbol 12-myristate 13-acetate (PMA)-induced COX-2 expression in both cancerous and non-cancerous breast cells. This effect was mediated by transcriptional inhibition of this gene [36]. Similarly, CXC chemokine receptor-4 (CXCR4) is expressed in various tumors. This receptor mediates homing of tumor cells to specific organs expressing the ligand CXCL12 and plays an important role in tumor growth, invasion, metastasis, and angiogenesis. Butein downregulated the expression of CXCR4. This effect correlated with the inhibition of CXCL12-induced migration and invasion of breast cancer cells. However, the decrease in CXCR4 expression induced by butein was not cell type-specific, and also occurred in pancreatic, prostate, multiple myeloma, head and neck, and hepatocellular cancer cell lines [37]. Controversial effects of butein on ROS production in breast cancer cells have been observed. An increase in ROS production was described in MDA-MB-231 (TNBC) cells. This resulted in decreased phosphorylation of extracellular signal-regulated kinase (ERK), increased p38 activity, diminished Bcl-2 expression, induced caspase 3 cleavage and was associated with poly ADP-ribose polymerase (PARP) cleavage and inhibition of breast cancer cell proliferation [38]. Two years later, Cho et al. demonstrated that butein inhibited ROS production with subsequent breast cancer growth suppression. These experiments were performed both in vitro and in vivo with butein-sensitive or -resistant breast cancer cells. Butein reduced the viabilities of different breast cancer cells, while not affecting resistant cells HER2 positive HCC-1419 (human cell line isolated from primary ductal carcinoma), SKBR-3 (human breast cancer cell line that overexpresses the Her2 gene product) and HCC-2218 cells (epithelial fibroblast cell isolated from the mammary gland ductal carcinoma). This was attributed to butein mediated inhibition of ROS levels resulting in the inhibition of AKT phosphorylation, as ROS regulate AKT activity and vice versa. The authors concluded that butein suppression of breast cancer growth was correlated with its reduction of the levels of both ROS and phosphorylated AKT, in vivo [39]. Additional experiments revealed that butein binds to a specific pocket of estrogen receptor (ERα) and promotes proteasome-mediated degradation of this receptor. Degradation results in butein-induced ERα downregulation, cell cycle arrest and inhibition of the growth of ERα+ breast cancer cells both in vitro and in vivo [40].

Indirect actions of butein in the modulation of breast cancer cell proliferation are represented by its inhibitory effect on estrogen production by aromatase inhibition apart of malignant breast cancer cells. Butein was found to be the strongest aromatase inhibitor among 5 tested hydroxychalcones with its IC_50_ value of 3.75 μM [41]. Another indirect effect appeared when breast cancer cells were co-cultured with fibroblasts, believed to play an important role in promoting the growth of breast cancer cells. Butein was found again as the most potent inhibitor of clonogenic growth of UACC-812 breast cancer cells isolated from the mammary gland ductal carcinoma. However, only when they were co-cultured with butein-pretreated fibroblasts [42]. Butein also reduced cell proliferation rate and the release of proinflammatory cytokines in two TNBC cell lines. However, cytokine release was suppressed only in MDA-MB-231 (Caucasian), and not in MDA-MB-468 (African American) cells, indicating different responses to butein treatment [40]. Similar results were also obtained in cardamonin evaluation in the same breast cancer cell lines (see below). Sensitive MDA-MB-231 breast cancer cells were then used to test the combination of butein with frondoside-A (triterpenoid glycoside from the Atlantic Sea Cucumber). The tests were performed both in vitro and in vivo CAM assay (on chick embryo chorioallantoic membrane) and revealed that butein alone significantly reduced cancer cell viability and colony growth, as well as migration and invasion. The effects were due to the potent inhibition of STAT3 phosphorylation (signal transducer and activator of transcription 3), leading to PARP cleavage and consequent cell death. Combination of butein with frondoside-A led to additive effects [43]. The antiproliferative and anticancer effects of xanthohumol and butein are summarized in Appendix A (see Appendix A).

### 2.3. Isoliquiritigenin

Isoliquiritigenin (2′,4′,4-trihydroxychalcone, ISL), a natural chalcone originally isolated from licorice root, has been found to have a broad spectrum of biological activities including anticancer activity [44]. Among other effects, it exhibits anti-proliferative, anti-angiogenic, and anti-invasive effects in breast cancer cells. It also acts as an estrogenic agonist of both ER isoforms and shows a dual effect in different cancer cells at different concentrations. Growth promotion is typical for low and intermediate ISL concentrations in hormone-sensitive MCF7 breast cancer cells. On the other hand, at a high level, ISL becomes cytotoxic in both ER-receptor positive as well as negative cells [45]. Antitumorigenic effects of ISL are mediated by different mechanisms and affect several cellular processes. Diminished cell viability, 5-bromo-2′-deoxyuridine (BrdU) incorporation, clonogenic ability and subsequent apoptosis were demonstrated in both MCF-7 and MDA-MB-231 cells breast cancer cells. Among others, these growth inhibitory effects and apoptosis were attributed to the downregulation of arachidonic acid metabolic network and the deactivation of the phosphatidylinositol 3-kinase/protein kinase B (PI3K/Akt) signaling pathway, which is considered one of the most important pathways regulating cell proliferation, the cell cycle and apoptosis. The results were supported by the finding that ISL inhibited mRNA expression of multiple forms of arachidonic acid-metabolizing enzymes as well as downregulated the levels of several phosphorylated forms of cell cycle regulating proteins [46]. An additional mechanism of cancer cell growth inhibition could be the ISL-induced block of β-catenin transcription activity. This mechanism, which significantly limits the side population and cancer stem cell (CSC) ratios in breast cancer cells, could have a synergistic effect with chemotherapeutic drugs. It could inhibit breast cancer cell proliferation and colony formation and enhance breast CSC chemosensitivity with little toxicity in normal tissues and mammary stem cells [47].

It is well accepted that microRNAs (miRNA) regulate tumor progression by modulation of the expression of several oncogenes and tumor suppressor genes. The effect of ISL on miRNAs’ regulation of tumorigenesis in breast cancer was evaluated in subsequent papers. The results revealed that ISL downregulated miR-374a. This is one of the main miRNAs involved in the migration and invasion of cancer cells. A decrease in miR-374a resulted in increased tumor suppressor gene PTEN expression (phosphatase and tensin homolog—tumor suppressor), and an inhibition of aberrant Akt signaling [48]. On the other hand, ISL increased miR-200c in BT-549 (papillary, invasive ductal tumor) and MDA-MB-231 TNBC cells. The effect correlated with subsequent ISL–induced inhibition of metastasis and tumor growth in nude mice models and was mediated by decreased c-Jun expression through the increase in miR-200c [49]. Several other mechanisms of ISL action in TNBC were discovered. ISL induced cell apoptosis, reduced Bcl-2 protein expression, increased Bax protein level and activated caspase-3 and PARP. Moreover, it reduced the expression of total and phosphorylated mammalian target of rapamycin (mTOR). All these effects resulted in the inhibition of TNBC MDA-MB-231 breast cancer cell growth through autophagy-mediated apoptosis [50].

Inhibition of angiogenesis is another possibility as to how to decrease cancer cell growth and proliferation. Some chalcones can inhibit neovascularization, influencing the effect of hypoxia-inducible factor-1a (HIF-1α), a signaling molecule in a central axis, activating oncogenic signaling. It also acts as a metabolic switch in endothelial cell (EC)-driven tumor angiogenesis. For example, ISL suppressed sprout formation from VEGF-treated aortic rings and inhibited cancer angiogenesis via significant inhibition of VEGF-receptor expression. This inhibition was mediated via the promotion of HIF-1a proteasome degradation [51].

The potential of ISL was also investigated as a preventive and therapeutic agent for breast cancer cell-induced metastatic bone destruction. ISL exerted its effect through the proteins directly involved in the differentiation, activity, and survival of osteoclasts. At non-toxic concentrations, it significantly inhibited the RANKL/OPG (receptor activator of nuclear factor kappa-B ligand/osteoprotegerin) ratio by reducing the production of RANKL and restoring OPG production in hFOB1.19 osteoblast cell culture stimulated with conditioned medium of MDA-MB-231 cells. ISL also reduced the expression of COX-2 in similarly stimulated hFOB1.19 cells. These effects may result in inhibitory potential of ISL on metastatic breast cancer-induced bone destruction [52].

### 2.4. Cardamonin

Cardamonin (CAR) is a chalcone isolated from *Alpiniae katsumadai*. It has been described as an anti-inflammatory and anti-tumor agent in breast, lung, colon, and gastric human cancer cell lines, in both in vitro culture systems as well as xenograft mouse models [53]. Anticancer activities were demonstrated in different breast cancer cells including drug resistant cancer stem cells [54], as well as in TNBC [55]. Cytotoxic effects of CAR in TNBC cells were attributed to the modulation of Bcl-2, Bax, cyt-C, cleavage of caspase-3 and PARP, with subsequent induction of apoptosis and cell cycle arrest. CAR also reversed epithelial–mesenchymal transition (EMT) and downregulated invasion and migration of BT-549 cells. In in vivo experiments, CAR significantly inhibited the tumor volume at dose of 5 mg/kg-treated mice [55]. Additional findings indicated that CAR inhibited the growth of the TNBC cell line MDA-MB-231 in vitro and in vivo by suppressing HIF-1α mediated cell metabolism. Suppression of HIF-1α enhanced mitochondrial oxidative phosphorylation and induced ROS accumulation [56]. The crucial role of ROS in the inhibition of CAR-induced breast cancer cell proliferation was confirmed after quenching of ROS by addition of N-acetyl-cysteine (NAC) or overexpression of catalase which also blocked CAR-induced cell cycle arrest and apoptosis [57]. An additional mechanism of antiproliferative action of CAR was observed in HeLa and MCF-7 breast cancer cells resistant to mTOR inhibitors. CAR overcame the resistance by mechanism of mTOR inhibition which differed from the currently available mTOR inhibitors. It decreased the expression of regulatory associated protein of mTOR (Raptor), the effect which resulted in inhibition of cell proliferation and decrease in phosphorylation of mTOR and S6K1 (ribosomal protein S6 kinase B1) in the mTOR inhibitor resistant cells [58]. Concerning drug-resistance, the effects of CAR were also evaluated in MDA-MB-231 (Caucasian) and MDA-MB-468 (African American) breast cancer cells. Both TNBC cell lines upregulated the expression of PD-L1, known to inhibit immune system control of cancer cell proliferation. CAR treatment caused a dose-dependent decrease in cell viability in both cell lines and downregulated PD-L1 expression in MDA-MB-231 cells. In MDA-MB-468 cells, CAR had an opposite effect, upregulating the expression of PD-L1 [59]. Besides having a direct effect on tumor cell proliferation, miRNAs also modulate the expression of genes that regulate tumor angiogenesis. These miRNAs have aberrant expression profiles in many different cancers, including breast cancers. CAR and some other phytochemicals were found to exhibit their anti-angiogenic properties by targeting the miRNAs that regulate EC metabolism [60].

### 2.5. Licochalcone A

Licochalcone A (LCA) (3-dimethylallyl-4,4′-dihydroxy-6-methoxychalcone) is a phytoestrogen extracted from licorice root. It expresses antioxidant, antibacterial, antiviral, antiparasitic and antitumor activities [61,62,63]. Former testing in several cell lines demonstrated that LCA induced apoptosis in the MCF-7 cell line. It decreased the anti-apoptotic protein Bcl-2 and altered the Bcl-2/Bax ratio in favor of apoptosis [64]. Additionally, it reduced the expression of cyclin D1 and promoted the cleavage of PARP. These effects preceded the cell cycle arrest at the G1 phase and induced apoptosis mediated by the intrinsic pathway [65]. Additional evaluation of the anticancer effects of LCA in MCF-7 cells revealed that the agent significantly decreased cell viability and promoted autophagy and apoptosis. The actions were attributed to the suppression of PI3K/Akt/mTOR signaling pathway [66]. Similar effects of LCA were described in MDA-MB-231 cells. It inhibited cell proliferation and cell cycle, modulated mitochondrial membrane potential and DNA damage, and reduced oxidative stress. It also activated cleaved-caspase 3 and 9, significantly decreased Bcl-2 expression and ultimately caused the release of cytochrome c from the mitochondria into the cytoplasm [67]. Comparing MCF-7 and MDA-MB-231 cells, LCA showed dysfunction of mitochondrial membrane potential and mitochondrial ROS production in both cell lines. Subsequent anti-proliferative and apoptotic effects passed the intracellular mitochondrial apoptosis pathway through regulation of Sp1 transcription factor and apoptosis-related proteins [62].

Moreover, it has been documented that LCA significantly suppresses breast cancer cells migration and invasivity. The effect of LCA on breast cancer cell invasion and migration was investigated in MDA-MB-231 cells. It was demonstrated that LCA effectively suppressed cell migration and invasion, and modulated E-cadherin and vimentin expression by blocking MAPK and AKT signaling [67]. Furthermore, LCA downregulated important genes associated with cancer development in MCF-7 and BT-20 breast cancer cell lines, including the AURKA protein (a member of a family of mitotic serine/threonine kinases). Moreover, they inhibited cell migration of metastatic BT-20 cells derived from a TNBC invasive ductal carcinoma. Reduction of MDR-1 protein by LCA was also observed. These findings indicate the anti-cancer, anti-metastatic and anti-resistance potential of both chalcones [68]. Recently, Gong et al. reported that protein arginine methyltransferases (PRMT) could be another molecular target of LCA. PRMT mediate arginine methylation implicated in multiple biological functions including transcriptional regulation. These enzymes were found to be upregulated in various cancers. Expression of PRMT6 is upregulated in human breast cancers and it is associated with oncogenesis. Studies of LCA effects in MCF-7 cells showed that it reversibly and selective inhibited PRMT6. This inhibition resulted in cytotoxicity by upregulating p53 expression and blocking cell cycle progression at a G2/M phase, followed by apoptosis. However, the agent was not effective towards non-cancer MCF-10A human breast epithelial cells [69].

### 2.6. Flavokawains

Flavokawain A (FKA) is a chalcone isolated from the root extracts of the kava-kava plant (*Piper methysticum*), belonging to the *Piperaceae* family. The extract is traditionally known as the Pacific elixir by the Pacific islanders for its role in a wide range of biological activities [70]. The anti-cancer properties of FKA in MCF-7 and MDA-MB231 cells were evaluated by measuring several parameters, indicating both apoptotic and metastatic effects. It induces apoptosis in both cell lines in a dose dependent manner through the intrinsic mitochondrial pathway. The migration and invasion process in MDA-MB231 was also inhibited. Similar effects were seen in the inhibition of the angiogenesis process performed in human umbilical vein endothelial cells (HUVECs) via tube formation assay and ex vivo rat aortic ring assay [71]. In addition, it has been shown that FKA preferentially reduces the viability of HER2-overexpressing breast cancer cell lines. Applied at cytotoxic concentrations to breast cancer cell lines, it had a minimal effect on the growth of non-malignant breast epithelial MCF10A cells. In combination with monoclonal anti-HER2 antibody trastuzumab, FKA enhanced its growth inhibitory effect and downregulated several transcription factors and cell cycle regulating proteins [72].

The immunomodulatory effects and the anti-inflammatory effects of FKA in a breast cancer murine model demonstrated an increased number of T cells (both Th1 cells and cytotoxic T lymphocytes) and elevated levels of IFN-γ and IL-2 in the serum. In the same time, FKA-treated mice had reduced levels of major pro-inflammatory mediators and decreased weight and volume of the tumor caused by apoptosis induction. All these effects indicated enhancement of antitumor immunity and prevention of inflammatory process in tumor microenvironment [73].

Similar in vivo antitumor and antimetastatic effects were also described in flavokawain B (FKB)-treated mice. The results were obtained in 4T1 (highly tumorigenic and invasive tumor cell line) tumors in mice. FKB regulated the immune system by increasing both helper and cytolytic T-cell and natural killer cell populations. It also enhanced the levels of IL-2 and IFN-γ, induced apoptosis and inhibited metastasis [73].

### 2.7. Garcinol

Garcinol, a polyisoprenylated chalcone containing two aromatic rings separated by a carbonyl group, is a natural agent extracted from the rind of the fruit of *Garcinia indica* (known as mangosteen) [74]. Several experiments have suggested its anti-cancer activity in breast cancer cells. Recently, a review article aimed to analyze the potential of *Garcinia* phytochemicals as a molecular therapy of breast cancer, evaluated the results of 28 article selected studies. The analysis showed that phytochemicals of *Garcinia*, including garcinol, have anti-cancer properties, resulting from apoptosis, inhibition of proliferation, and metastasis in breast cancer cells [75]. The mechanisms involved in its anti-cancer effects included the reversal of EMT associated with the upregulation of epithelial marker E-cadherin and downregulation of some mesenchymal markers including vimentin, ZEB1 and ZEB2. In addition, garcinol upregulated the expression of miR-200 and let-7 family miRNAs participating in EMT. Transfection of cells with NF-κB p65 subunit and anti-miR-200s attenuated the effect of garcinol on both apoptosis induction and breast cancer cell invasion. The results were confirmed in in vivo xenograft mouse model studies, where garcinol inhibited NF-κB, miRNAs, vimentin, and nuclear β-catenin [76]. Furthermore, garcinol-induced inhibition of acetyltransferase and the effect on cell proliferation, cell cycle progression and apoptosis were investigated in estrogen-stimulated MCF-7 cells. Treatment with garcinol resulted in the inhibition of proliferation, cell cycle progression arrest at the G0/G1 phase, and the increase in apoptosis. These effects were linked to hyperacetylation levels of histones and nonhistone NF-κB/p65 [77].

### 2.8. Isobavachalcone

Isobavachalcone (IBC) is a naturally occurring prenylated chalcone derived from the seeds of *Psoralea corylifolia *L. It is known as a phytoestrogen with an antitumor effect. IBC concentration- and time-dependently induced apoptosis of both MCF-7 and MDA-MB-231 cells. It induced inhibition of MCF-7 cell proliferation, triggered apoptosis and autophagy. These effects were mediated by increasing Bax expression and downregulating the expressions of Bcl-2, Akt and p-Akt-473 proteins [78]. In MDA-MB-231 cells, IBC action resulted in multiple cell death processes (apoptosis, necroptosis, autophagy). These were attributed to the downregulation of Akt and p-Akt-473 and an increase in the Bax/Bcl-2 ratio. In addition, IBC induced mitochondrial dysfunction, thereby decreasing cellular ATP levels and increasing ROS accumulation [79]. IBC may also interfere with ERα and influence estradiol-induced paclitaxel resistance. It downregulated ERα. This effect resulted in decreased expression of CD44 (cell-surface glycoprotein involved in cell–cell interactions) and thus inhibited tumor growth of ER-positive breast cancer cells in paclitaxel-resistant xenograft models [80].

### 2.9. Other Chalcones

Panduratin A is isolated from plants like *Boesenbergia pandurata.* The apple polyphenols phloretin and pinostrobin are from *Uvaria chamae*, and some other plants possess many health benefits. Treatment of MCF-7 breast cancer cells with panduratin A resulted in a time- and dose-dependent inhibition of cell growth. The mechanism resulted from increased activity/expression of mitochondrial cytochrome C, caspases 7, 8 and 9 with a significant increase in the Bax/Bcl-2 ratio, suggesting the involvement of a mitochondrial-dependent apoptotic pathway. Several other proteins involved in cell cycle regulation were also affected [81]. Evaluation of pinostrobin resulted in selective inhibition of the migration of both MDA-MB-231 and T47D malignant cells, while the effects on MCF10A cells were blunted. Surprisingly, the inhibitory actions on cell adhesion, cell spreading, and focal adhesion formation of both malignant cell lines were not accompanied with anti-proliferation effects [82]. A distinct mechanism of action inhibiting cell proliferation of MDA-MB-231 TNBC cell line was discovered in phloretin. This apple polyphenol, which is specific antagonist of GLUT2 protein, inhibited MDA-MB-231 cell growth and arrested the cell cycle. It also decreased migration of the MDA-MB-231 cells through the inhibition of paxillin/FAK, Src, and alpha smooth muscle actin (α-sMA) through the activation of E-cadherin. In addition, the anti-tumorigenic effect of phloretin was demonstrated in vivo using BALB/c nude mice bearing MDA-MB-231 tumor xenografts. The authors concluded that the inhibition of GLUT2 phloretin could potentially suppress TNBC tumor cell growth and metastasis [83]. Similarly, a totally different mechanism of cell death in breast cancer cells was ascribed to *trans*-chalcone. In breast cancer cells, its anti-proliferative effects were considered to be mediated through the increase in heme oxygenase-1 expression. Blocking this enzyme diminished the effect of *trans*-chalcone on cell growth inhibition [84].

## 3. Chalcones and Cancers of the Digestive System

Digestive system cancers, including colorectal, stomach, liver, pancreas and esophagus, represent more than 35% of all deaths attributed to cancer in 2020. Among them, colorectal and stomach cancers belong to the most common cancers of digestive system with more than 2.9 million of new cases and over 1,700,000 deaths in 2020 [2]. Because antiproliferative and anticancer effects of chalcones related to colorectal and gastric cancers have been reviewed recently [85], in this paper we summarize evidence about the antiproliferative effect of natural chalcones in cancers of liver, pancreas, mouth and esophagus.

### 3.1. Liver Cancers

In 2020, more than 900,000 new cases of liver cancer were diagnosed, and over 830,000 patients died from this type of cancer. It is predicted that mortality of liver cancer will increase by 56.4% in 2040 [86]. Hepatitis B or C infections, chronic alcohol consumption, and non-alcoholic fatty liver disease are the primary etiological factors associated with the risk of liver cancer [87]. In addition, around 90% of liver cancer cases are attributed to hepatocellular carcinoma (HCC), which is the most prevalent type of liver cancer. Because chemotherapy is generally insufficient, the prognosis of liver cancer is poor [88]. On the other hand, preclinical studies with natural compounds have shown their potential to inhibit liver cancer progression [89].

#### 3.1.1. Xanthohumol

Among other things, the antiproliferative effect of XH has also been studied in HCC cell lines. Exposition of HepG2 and Huh7 cell lines to XH at a dose of 25 μM led to induction of apoptosis associated with caspase-3 activation. Moreover, significant decrease in cancer cell proliferation and migration has also been documented. In contrast, no effect on viability was observed in XH-treated primary human hepatocytes [90]. Proapoptotic and antiproliferative effects of XH on HCC cancer cell lines with reduced antiprolifrative activity against non-cancer liver cells have been also documented by Ho et al. [91]. Another study showed that antiproliferative and proapoptotic effect can be associated with ability of XH to inhibit the Notch signaling pathway [92]. The authors of this study revealed that exposition of HCC cell lines (Huh-7, HepG2, Hep3B, and SK-Hep-1) to XH significantly reduced cell viability, proliferation, clonogenicity and induced apoptosis. All these effects were associated with suppression of Notch signaling as evidenced by the downregulation of Notch1 and HES-1 proteins. Later, Zhao et al. [93] documented, that cytotoxic and antiproliferative effects of XH in HCC cell line HepG2 is mediated via NF-κB/p53 signaling pathway. XH suppressed NF-κB expression and initiated p53 expression. This resulted in increased expression of proapoptotic proteins and vice versa, decreased expression of antiapoptotic proteins. Cancer cell selectivity and ability to induce apoptosis in HCC cell line HepG2 was also confirmed recently [94]. In addition, in vivo chemopreventive effect of XH has also been observed. Consumption of drinking water supplemented with XH significantly reduced the number of liver preneoplastic lesions in rats induced by application heterocyclic aromatic amine [95].

#### 3.1.2. Butein

In HCC cell lines, butein suppressed proliferation via cell cycle arrest at the G2/M phase and apoptosis induction. The following analyses showed activation of DNA damage response system and checkpoint kinase 1 (Chk1) and Chk2 phosphorylations. Moreover, increased ROS generation in butein-treated HCC cells was observed. It seems that increased ROS are involved in antiproliferative effect of butein because co-administration of NAC or glutathione significantly lowered butein-induced G2/M arrest and apoptosis [96]. Moreover, in TNF-related apoptosis-inducing ligand (TRAIL)-resistant HCC cells butein augmented TRAIL-induced apoptosis probably due to increased DR5 expression [97].

It is well known that HCC carries great potential to metastasize in extrahepatic tissues including lungs, lymph nodes or bones [98]. One of the first steps of metastasis is cell invasion [99]. Butein has been documented to inhibit invasion and migration of the HCC cell line SK-Hep-1 in non-toxic concentration. Deeper analysis showed that butein inhibited either activity or protein expression of several proteolytic enzymes involved in extracellular matrix degradation, including MMP-2, -7, -9, uPA (urokinase-type plasminogen activator) as well as several other proteins involved in cell migration and invasion such as Ras, Rho A and ROCK1 (a serine/threonine kinase). In addition, it is suggested that these activities are closely related to the modulation of ERK1/2, JNK1/2, p-p38, and p-c-Jun signaling pathways in butein-treated HCC cells [100]. The antimetastatic potential of butein was also confirmed by Liu et al. [101] who found that butein inhibited Akt/mTOR/p70S6K signaling with subsequent inhibition of MMP-9 and uPA activity in SK-Hep-1 HCC cell line. Moreover, marked suppression of cancer metastasis has been demonstrated also in vivo in CAM assay. Additionally, some authors documented the anticancer effect of butein in nude mice bearing HCC xenograft. Butein significantly restrained the growth of tumors and the anticancer effect of butein was associated with inhibition of either Aurora B kinase activity [102] or by suppression of the STAT3 signaling pathway [103].

#### 3.1.3. Cardamonin

In HCC HepG2 cells, CAR has been found to inhibit cancer cell proliferation and to induce both extrinsic and intrinsic pathways of apoptosis at IC_50_ = 17.1 μM. These effects were associated with intracellular ROS generation and inhibition of the NF-κB pathway [104]. Dysregulation of NF-κB pathway has also been reported in vivo in athymic nude mice bearing HCC xenograft. Oral administration of CAR significantly suppressed tumor growth and immunohistochemical analyses showed low levels of proliferative proteins, including PCNA (proliferating cell nuclear antigen) and Ki-67. In addition, decrease in antiapoptotic Bcl-2 protein and increase in proapoptotic Bax protein has also been recorded. These effects were accompanied with downregulation of NF-κB-p65, and Ikkβ proteins suggesting that anticancer effect of CAR may be mediated through NF-κB pathway [105].

#### 3.1.4. Licochalcones

Among numerous active phytochemicals, LCA and licochalcone B (LCB) have been reported to have antiproliferative effects in HCC cell lines. In subtoxic doses (5–20 μM), LCA inhibited migration and invasion of human HCC cells. These effects were accompanied by downregulated expression of uPA as well as uPA mRNA. Moreover, zymography analysis revealed also decreased function of uPA. As uPA is involved in the degradation of extracellular matrix, results of this study indicate anti-metastatic potential of LCA [106]. The study by Wang and co-workers revealed that LCA in higher doses suppressed HCC cell proliferation and induced apoptosis [107]. Detailed analyses showed that LCA modulated expression of several genes related to apoptosis or cell cycle such as survivin, cyclin B1, and CDK1 or different caspases and Bcl-2 family proteins. In addition, another study documented the dysregulation of numerous miRNAs in HepG2 cells exposed to LCA. Of them, miRNAs related to the FOXO signaling pathway were significantly enriched [108]. Furthermore, in 2018, Wu et al. [109] found that the combination of sorafenib (a multikinase inhibitor used for the treatment of HCC) and LCA led to a significant reduction in the invasion and migration of HCC cells in vitro, and also suppressed the HCC cell-mediated lung metastasis in vivo.

Similar to LCA, LCB has also been referred as a potent suppressor of HCC viability and proliferation. It inhibited growth of HepG2 cells with IC_50_ = 110.15 μM. Performed analyses revealed cell cycle arrest at G2/M phase, apoptosis induction and increased ROS generation. Additionally, both extrinsic and intrinsic apoptotic pathways were activated [110]. Regarding molecular mechanism of antiproliferative action of LCB, a pleiotropic mode of action was reported. Zhang et al. [111] reported antiproliferative and proapoptotic effect of LCB in HepG2 and HuH7 HCC cells. These effects were accompanied with DNA damage, cell cycle arrest at G2/M phase, PARP cleavage, caspase-3 activation and upregulation of DR4 and DR5. Moreover, LCB also modulated activity of several signaling pathways, including Akt/mTOR pathway (inhibition) or MAPK pathways (activation). Furthermore, LCB has been found to either upregulate or downregulate more than 760 miRNAs [112]. Another study documented anticancer effects of LCB in vivo in diethylnitrosamine (DEN)-induced hepatocarcinoma in rats. LCB alone or in combination with fullerene nanoparticles prevented DEN-induced histopathological changes in treated rats. Further analyses showed that LCB/fullerene combination attenuated DEN-induced oxidative DNA damage and DNA fragmentation and antiapoptotic proteins expression with the current increase in proapoptotic proteins expression [113]. The antiproliferative and anticancer effects of cardamonin and licochalcones are summarized in Appendix A. (see Appendix A).

#### 3.1.5. Other Chalcones

In addition to the above-mentioned chalcones, many other ones have been documented to repress liver cancer cell growth in vitro or in vivo. Hydroxysafflor yellow A (HSYA) decreased HepG2 cells viability via induction of autophagy [114]. However, a detailed study by Wu et al. [115] showed that although HSYA initiated the first phase of autophagy, it blocked the late phase of autophagic flux with subsequent induction of apoptosis. Another study showed that millepachine induced cell cycle arrest at the G2/M phase followed by apoptosis in HCC cells in vitro and significantly suppressed tumor growth in vivo in nude mice [116]. Later, Yang and co-workers [117] uncovered that millepachine-induced G2/M arrest can be the result of its interaction with the colchicine-binding site in β-tubulin. Furthermore, several other reports indicated that chalcones such as IBC [118], 4-hydroxyderricin [119], broussochalcone A [120], phloretin [121], ISL [122,123], bavachalcone [124] or flavokawains [125] can also be considered as potential therapeutic agents for the treatment of HCC.

### 3.2. Pancreatic Cancer

In developed countries, pancreatic cancer is among the leading causes of cancer-related deaths and is considered one of the deadliest types of cancer globally. According to GLOBOCAN 2020, pancreatic cancer results in over 466,000 deaths and it ranks as the seventh most common cause of cancer-related deaths in both sexes [2]. Despite advances in multimodality treatment involving surgery and adjuvant therapy, the mortality rate of pancreatic cancer continues to rise over time [126] and therefore it is important to develop new drugs for pancreatic cancer treatment.

#### 3.2.1. Xanthohumol

Among chalcones, XH has been referred to suppress the growth of several pancreatic cell lines including AsPC-1, PANC-1, L3.6pl and MiaPaCa-2 as well as patient-derived pancreatic cancer cells named 512 and 651. In addition, XH significantly decreased ability of pancreatic cells to form colonies. Antiproliferative effect of XH was associated with apoptosis induction via modulation of Notch1 signaling pathway [127]. Some other authors [128] reported that XH in pancreatic cell lines inhibited STAT3 phosphorylation and expression of genes involved in apoptosis and cell cycle control including *cyclinD1*, *survivin*, and *Bcl-xL*. Furthermore, in a xenografted mice model, intraperitoneal administration significantly decreased the volume and weight of tumors in XH-treated nude mice. In a subsequent study, Saito and colleagues [129] discovered that even at low concentrations XH interfered with the angiogenesis in pancreatic cancers by blocking the NF-κB signaling pathway, both in vitro and in vivo. Among others, XH inhibited the expression of vascular endothelial growth factor (VEGF), interleukin-8 or tube formation by HUVECs. Suppression of the NF-κB signaling pathway has also been documented recently by Krajka-Kuźniak et al. [130] in XH-treated pancreatic cell line PANC-1.

#### 3.2.2. Other Chalcones

In addition, some other chalcones have also been reported to suppress pancreatic cell proliferation or migration via dysregulation of NF-κB signaling. Chua et al. [37] observed that butein significantly downregulated the expression of CXCR4, which plays a crucial role in processes such as angiogenesis, tumor proliferation invasion or metastasis. Subsequent analyses showed that butein-induced CXCR4 downregulation was the result of NF-κB inhibition. Later, Parasramka and Gupta [131] reported that garcinol significantly decreased pancreatic cell lines proliferation via downregulation of nuclear transcription factor NF-κB with subsequent induction of apoptosis, reduction of MMP-9 activity, inhibition of VEGF and IL-8 production. Another study showed that garcinol markedly suppressed survival and migration of pancreatic cancer cell line BxPC-3 due to the inhibition of expression and transcriptional activity of STAT-3 [132]. In addition to the modulation of NF-κB and STAT3 signaling, several other signaling pathways associated with cell survival and death have been referred to as target molecules for chalcones, including p38 MAPK signaling [78], FOXO3a-FOXM1 Axis [133], JNK pathway [134] or NR4A1-dependent apoptotic pathway [135].

### 3.3. Oral Cancers

Cancers of the oral cavity (including the lips) are very frequent in countries of South-Central Asia and Melanesia. More than 377,000 of new cases have been estimated worldwide in 2020 [2]. Among oral cancers, oral squamous cell carcinoma (OSCC) is the most common one. There are many risk factors including tobacco chewing and smoking, betel nut, alcohol drinking, HPV virus infection and several others [136]. Moreover, despite the significant improvement in clinical treatments for oral cancer over the past decade, the 5-year survival rate of patients with oral cancer has only been marginally improved. Among different natural phytochemicals, chalcones have also been reported to suppress the proliferation, invasion or migration of OSCC cells.

#### 3.3.1. Licochalcones

Licochalcones, isolated from the roots of *Glycyrrhiza inflata*, have been examined as perspective compounds for OSCC treatment. Kim et al. [137] found that LCA induced extrinsic apoptosis pathway in KB (human oral cancer cells) cells as documented by increased expression of FasL at both RNA and protein levels. This effect has been associated with increased phosphorylation and activation of ERK and p38, the members of MAPK signaling. Another study showed that LCA suppressed growth and induced apoptosis in OSCC cells due to the downregulation of transcription factor SP1 (specificity protein 1) expression and its downstream proteins [138]. Similar effect i.e., induction of apoptosis in OSCC cells exposed to LCA, has been reported by Zeng et al. [139]. In SSC-25 oral carcinoma cells, LCA induced both extrinsic and intrinsic apoptotic pathways associated with activation of caspase-8 and -9, PARP cleavage and DNA fragmentation. Later, Hao et al. [140] demonstrated the association of apoptosis induction and inhibition of SCC4 oral carcinoma cell migration with a significant reduction of PI3K/AKT pathway and ROS production. Moreover, the anticancer effect of LCA has also been documented in vivo in nude mice bearing SCC4 cells. In addition, several other licochalcones have also been studied for their potential to suppress the survival and proliferation of oral carcinoma cells. Licochalcones B, C, D and H induced apoptosis in OSCC cells via different mechanisms including ROS generation, inhibition of the JAK2/STAT3 signaling pathway with downregulation of genes such as surviving, p21 and p27, or via regulation of matrin 3, a protein involved in several cell processes such as apoptosis, DNA replication, transcription and translation [141,142,143,144,145].

#### 3.3.2. Other Chalcones

Numerous other chalcones have demonstrated inhibitory activity against OSCC cells via their modulatory potential against different targets. Bordoloi et al. [146] found that butein suppressed OSCC survival due to downregulated expression of NF-κB and its downstream molecules such as COX-2, survivin or MMP-9. Another author documented tumor-suppressive effect of ISL in OSCC cells mediated by the downregulation of DNA repair mechanism resulting in DNA damage and apoptosis induction [147] or via inhibition of OSCC cancer stem cells accompanied by the regulation GRP78 chaperone expression [148]. Furthermore, Li et al. [149] in their study discovered that the antiproliferative effect of XH is closely related to survivin destruction due to increased ubiquitination of this antiapoptotic protein.

### 3.4. Esophageal Cancers

In 2020, esophageal cancer was the cause of 544,000 deaths, placing it sixth in terms of mortality. In addition, among Bangladeshi men and women and Malawian men, esophageal cancer is even the top cause of cancer-related deaths [2]. Current therapy includes surgery, radiotherapy and chemotherapy. However, due to limited efficacy and severe toxicity of traditional therapy, the outcomes are still unsatisfactory, and there is a need for a new, more safe therapy. Among several phytochemicals, natural chalcones have been reported to affect the survival and proliferation of esophageal cancer cell lines.

#### 3.4.1. Licochalcones

Licochalcone C (LCC) suppresses the viability of esophageal squamous cell carcinoma (ESCC) cells in a dose-dependent manner. Moreover, it promotes apoptosis through increased generation of ROS with subsequent activation of MAPK pathway. In addition, further analyses showed block of cell cycle at G1 phase in ESCC cells exposed to LCC associated with cyclin D1 downregulation and upregulation of p21 and p27 proteins [150]. Another study showed that LCB caused G2/M phase cell cycle arrest in ESCC cells associated with downregulation of cyclin B1 expression. Detailed analysis showed that LCB significantly inhibited JAK2 kinase activity followed by a decrease in STAT3 phosphorylation [151]. A recent study demonstrated that another licochalcone, licochalcone H (LCH), induced cell cycle arrest at the G2/M phase via modulation of several proteins associated with the cell cycle including cdc2, cyclin B1, p21, and p27. This study also approved that LCH induced apoptosis via triggering of endoplasmatic reticulum stress accompanied and activation of JNK/p38 pathway in ESCC cells [152].

#### 3.4.2. Other Chalcones

In addition, several other chalcones have been reported to affect ESCC cell survival via pleiotropic mechanisms of action. Phlorizin, extracted from sweat tee, has been found to inhibit cell proliferation, invasion and migration through suppression of JAK/STAT3 signaling and autophagy inhibition [153]. Another study showed that liquiritigenin from licorice stopped the progression of cell cycle at the G2/M phase and downregulated cyclin B1 and cdc2 expression in ESCC cells. Moreover, it stimulated ROS production and modulated mitochondrial function as evidenced by the loss of mitochondrial membrane potential and by changes in the expression of Bcl-2 family proteins [154]. These authors obtained similar results also with echinatin, another chalcone from licorice. Echinatin in ESCC cells induced arrest at the G2/M phase of the cell cycle associated with cyclin B1, and cdc2 downregulation with the current upregulation of p21 and p27 proteins. In addition, cultivation of cells with echinatin stimulated ROS production with subsequent induction of apoptosis mediated via ER stress and p38 MAPK/JNK activation [155]. Later, they described similar antiproliferative mechanisms also in 3-deoxysappanchalcone-treated ESCC cells [156]. Studies by Liu et al. [157] and Yin et al. [158] described XH as a perspective agent against ESCC both in vitro and in vivo. In the first study, G1 block was observed in XH-treated ESCC cells. Moreover, activation of caspase-3 and -7, cleavage of PARP, release of cytochrome *c* as well as upregulation of Bax and Bim. It was suggested that these effects were related to inhibition of AKT kinase activity followed by reduced phosphorylation of downstream proteins. In addition, XH significantly reduced tumor volume and weight in mice bearing patients-derived xenografts (PDX) having overexpressed AKT. The authors of the second study discovered that XH inhibited ESCC cell proliferation and induced apoptosis by targeting keratin-18 (KRT18), which plays a crucial role in maintaining of tissue integrity and is often over-expressed in different cancer types. This mechanism is probably involved also in the anticancer effect of XH as it was confirmed by its ability to suppress the growth of PDX tumors with a high level of KRT18. Another author reported the antiproliferative and anticancer effects of garcinol. Incubation of ESCC cells with garcinol resulted in the inhibition of cancer cell proliferation and metastasis. Garcinol prevented lung metastasis in animals injected with ESCC cells. These effects were associated with the suppression of TGF-β1 signaling and decreased levels of p300 (a protein involved in cell proliferation, migration, and invasion) in the nucleus [159]. Another study [160] documented that CAR also possesses a strong potential to modulate esophageal cancer proliferation and metastasis in vitro and in vivo. The study showed the ability of CAR to reverse EMT, induce apoptosis and inhibit PI3K/AKT signaling pathway in ESCC cells. In addition, peroral administration of CAR showed a dose-dependent anticancer effect in mice bearing *s.c.* human ESCC cells xenograft. Moreover, immunohistochemical analyses showed reduced expression of phosphorylated PI3K and AKT which is in accordance with in vitro results.

## 4. Chalcones and Lung Cancer

Although the prevalence of lung cancer varies depending on the population, geographic region, and many other factors, according to data provided by the International Agency for Research on Cancer (IARC), lung cancer is the second most commonly diagnosed cancer (11.4%) after breast cancer (11.7%), and the leading cause of cancer death (18.0% of total cancer deaths) for both sexes. Approximately 2.2 million cases are diagnosed annually, with nearly 1.8 million cases resulting in death [2,161].

Despite the development of new electronic nicotine delivery systems (ENDS), which aim to provide a safer alternative to smoking, tobacco smoking remains the most significant and widespread risk factor for the development of lung cancer [162]. Occupational exposures (radon, asbestos, metals), ionizing radiation, air pollution, and other factors play a significant role in the etiology of lung cancer. Biological and genetic factors associated with lung cancer include chronic lung diseases, infections, positive family history, and various genetic alterations (such as mutations in the EGFR, KRAS, BRAF, ERRB2 genes, among others) [163,164]. There are close associations between the genetic profile and histology/morphology, the understanding of which is essential for the correct diagnosis and most appropriate treatment [165]. According to the current WHO classification from 2011, lung carcinomas are classified as adenocarcinoma (ADC), squamous cell carcinoma (SCC), small cell lung cancer (SCLC), large cell neuroendocrine carcinoma (LCNEC), non-small cell carcinoma-carcinoid tumor not otherwise specified (NSCC-NOS), etc. [166]. Although the mortality rate from lung cancer is decreasing over the years, the identification of “druggable” oncogenes (such as EGFR, ALK) provides the opportunity for the study and implementation of effective targeted therapy [167]. Among the potential molecules that exhibit pleiotropic and selective effects against tumor cells are chalcones [168,169]. We also present natural chalcones with demonstrated anti-tumor effects against lung tumors in in vivo and in vitro conditions.

### 4.1. Xanthohumol

Several studies have demonstrated the significant efficacy of XH against various lung cancer cells, including A549, H1563, H520, H357, H23, H1975, and HCC827, in a concentration and time-dependent manner [170,171,172,173].

Inhibition of proliferation and colony formation was observed at a concentration of 2 µmol/L after 72 h of exposure [170]. XH-induced DNA fragmentation led to the initiation of DNA repair mechanisms and cell cycle arrest. The p53 protein and its downstream proteins, inhibitors of cyclin/cyclin-dependent kinase (CDK) complexes p21 and p27, act as sensitive DNA damage sensors [174]. XH increased their levels in tumor cells, leading to cell cycle arrest in the G0/G1 and S phases, downregulation of cyclin D1, which is highly expressed in NSCLC cells, and necessary for maintaining tumorigenic properties of cancer cells [170,172,173]. In addition to cell cycle arrest, antiproliferative effects of XH also include the induction of apoptosis. Proapoptotic proteins from the Bcl-2 family participate in mitochondrial damage. PUMA (p53 upregulated modulator of apoptosis) is a “BH3-only” protein that triggers Bax/Bak mitochondrial membrane translocation and the propagation of apoptotic stimuli [175]. Li et al. [171] reported that upregulation of PUMA is necessary for the initiation of the mitochondrial pathway of apoptosis. The reduction of mitochondrial membrane potential with subsequent cytochrome *c* release contributed to the activation of caspases -3, -8, -9, and the externalization of phosphatidylserine, which is considered a marker of apoptosis [172,173]. Other studies have shown that the mechanism of XH antiproliferative effect involves influencing signaling pathways and transcription factors that regulate cell survival and growth, proliferation, cell cycle, metabolism, and cell death. A significant target for XH is the Ras/Raf/MEK/ERK pathway. XH-induced suppression of this pathway through the inhibition of phosphorylation of Erk1/2, p90RSK, and pCREB was comparable to the effect of U0126—a specific inhibitor of MEK1/2 kinases [172]. The inhibition of this pathway also depends on the inhibition of cyclin D and Fra1 expression, which is phosphorylated by Erk1/2. In HCC827 cells, XH supported the ubiquitination of Fra1, which promotes cell proliferation, cell survival, and angiogenesis in human tumors [170]. Furthermore, it is well known that overexpressed T-lymphokine-activated killer cell-originated protein kinase (TOPK) contributes to tumorigenesis, as well as invasiveness and metastasis of tumor cells and is associated with poor prognosis for patients with lung cancer [176]. Treatment with XH significantly reduced the phosphorylation of TOPK as well as its downstream signaling molecules Akt and histone H3 in A549 and HCC827 cells and also suppressed its kinase activity. Biomolecular interaction analysis showed that this effect is mediated by direct interaction between XH and TOPK [177]. The targets of XH also include topoisomerases, enzymes essential for DNA relaxation during replication, transcription, mitosis, and other processes. XH is able to inhibit the activity of TOPO I and significantly reduce the expression of ABCB1, ABCC1, ABCC2, and ABCC3, efflux drug transporters that are associated with resistance to anticancer chemotherapy [178,179]. Another study showed that anticancer effect of XH also includes modulation of the tumor microenvironment and angiogenesis. Sławińska-Brych et al. [180] documented stimulated migration, invasion, and expression of some EMT markers in phorbol-12-myristate-13-acetate (PMA)-treated A549 cells. XH significantly reduced the expression of MMP-2 and MMP-9, strongly associated with the metastatic potential of many tumor cells. On the other hand, it increased the level of TIMP-1, an endogenous tissue inhibitor of metalloproteinase proteins. XH also interfered with PMA-induced expression of VEGF and TGF-β, the main regulators of angiogenesis and EMT [181]. The inhibition of EMT by XH was also demonstrated by downregulating vimentin, N-cadherin, and Snail (mesenchymal markers) and increasing the levels of E-cadherin and α-catenin (epithelial markers) [182].

All of these results suggest the high anti-tumor potential of XH. This has also been demonstrated by experiments on animal models. Tumors from the A549, H358 and HCC827 xenograft mouse models showed significantly lower weight and volume after treatment with XH (10–25 mg/kg), without affecting the body weight of the mice or liver function. Immunohistochemical staining showed in vivo reduced expression of Ki-67, cyclin D1, and Fra1 with concurrent upregulation of the proapoptotic protein PUMA. All of the aforementioned results demonstrate the potential of XH as an anti-tumor agent for the management of lung cancer [170,171,177].

### 4.2. Butein

Due to its proven antiproliferative properties, it has also been studied in relation to lung carcinoma [39,183]. Butein is capable of inhibiting cell proliferation and reducing the viability of lung cancer cell lines at a concentration of 10 µmol/L. The colony growth assay can be used to evaluate the antiproliferative potential of interesting prospective molecules with anti-tumor effects. Treatment with butein at various concentrations (10–100 µmol/L) significantly reduced the number of colonies and proliferation ability. Cell cycle analysis of A549 cells incubated with the chalcone showed a significant increase in the population of cells in the G0/G1 phase of the cell cycle, and in concentrations of 20–60 µmol/L, the cell cycle of A549 and PC-9 cells was arrested in a concentration-dependent manner in the G2/M phase [43,184,185]. The inhibition of the cell cycle is the result of changes in the levels and activity of cell cycle regulating proteins. These include cyclin family proteins and cyclin-dependent kinases. The arrest of the cell cycle in the G2/M phase induced by butein is associated with a decrease in the levels of cyclin D, the phosphatase cdc25c, and the cyclin-dependent kinase cdc2. Di et al. [184] further investigated apoptotic cell death in lung cancer cells. The results of the TUNEL staining assay, which can identify DNA fragmentation, showed apoptosis present in cells treated with butein. This phenomenon was associated with a significant decrease in mitochondrial membrane potential as a result of changes in the levels of proteins from the Bcl-2 family. There was an upregulation of proapoptotic Bax and PUMA proteins and downregulation of Bcl-2. These results suggest the activation of the mitochondrial pathway of cell death. The endoplasmic reticulum (ER) may also be involved in the induction of apoptosis. Damage to the ER results in a reaction known as the unfolded protein response (UPR), which is a manifestation of ER stress and is accompanied by the activation of PERK (PRKR-like ER kinase), IRE1, and ATF6 sensors, which function as signal transducers. After incubation of cells with butein, there was an upregulation of markers downstream of these three proteins (phospho-eIF2α, CHOP, ATF4, XBP1), including increased levels of IRE1α and phosphorylated PERK [186]. The authors suggest that this mechanism is associated with oxidative stress (presence of ROS, increased activity of NADPH oxidase, decreased levels of GSH and SOD2). All of these processes result in the activation of executioner caspases. The involvement of caspases-3, -8, and -9 in butein-induced apoptotic cell death was demonstrated using the pan-caspase inhibitor z-VAD. One marker of apoptosis is the cleavage of the DNA repair enzyme PARP, which was identified in several lung cancer cell lines after treatment with butein [43,187]. Butein also inhibits signaling pathways associated with proliferation, survival, metastasis, invasiveness, and angiogenesis overactivated in lung tumors. These include the inhibition of COX-2 [185], the inhibition of the phosphorylation of STAT3, activated in many tumors [43], or the activation of the stress kinase p38, accompanied by oxidative stress, DNA damage, and mitochondrial damage [188]. A study on HCC827 (gefitinib-sensitive) and HCC827GR (gefitinib-resistant) NSCLC cells surprisingly showed that butein directly binds to EGFR and inhibits its phosphorylation. In addition, it suppresses MET kinase activity and prevents the interaction of MET with PI3K. Its antiproliferative effect also includes the downregulation of phosphorylated ERBB3, Akt, and Erk1/2. The changes in expression compared to the effects of gefitinib and PHA665752 (MET inhibitor) were nonsignificant [187].

Currently, in the research of antitumor drugs, emphasis is placed on finding molecular targets that could be used in immunotherapy. In the cells of many types of cancer, PD-L1 protein is massively expressed, which ensures the apoptosis of activated T cells and thus provides an escape for tumor cells from immune surveillance. Low-molecular-weight PD-L1 inhibitors have shown satisfactory results in clinical studies [189]. Butein reliably reduced the expression of PD-L1 in cells H292, H460, HCC827, four NSCLC lines (H1975, H1299, A549, and PC-9), and three primary cells derived from patients. Butein interacts with IFNγ-induced transcription in a time- and dose-dependent manner. The expression of PD-L1 is influenced by transcription factors such as STAT1, STAT3, and c-Myc. The authors believed that the changes in PD-L1 levels in butein-treated cells were involved in signaling associated with STAT1. It was demonstrated that in all tested lines, butein suppressed the expression of STAT1 and its phosphorylated form, thereby enhancing the killing ability of T cells [190].

The effects of butein in in vivo models are also remarkable. As mentioned above, the anticancer effect of butein is mediated by its ability to induce ER stress and significantly increase ROS generation. This effect was demonstrated in an experiment on PC-9 xenografted nude mice, where the application of NAC and 4-PBA (CHOP siRNA) significantly weakened butein’s inhibitory effect on tumor growth [184]. This effect at a dose of 5 mg/kg was also observed in comparison with gefitinib (5 mg/kg), where in HCC827-induced tumor cells, it induced a comparable effect, and in resistant HCC827GR tumor cells, unlike butein, gefitinib had no significant effect [187]. Similar results (50% inhibition of growth in the CT26 xenograft model) were reported after the administration of 20 mg/kg of butein. The role of butein in the regulation of PD-L1 is underlined by the fact that in immunodeficient mice, butein had no anticancer activity, unlike in immunocompetent mice, where a higher ratio of tumor-infiltrating CD8+ T cells was observed [190].

### 4.3. Isoliquiritigenin

The effect of ISL on tumor lung cells in vitro has been studied for many years. Several studies have demonstrated its ability to inhibit the growth and proliferation of various types of tumor cells, including lung cancer cells, either alone or as part of plant extracts [191]. In A549 cells (NSCLC), ISL is able to induce cell cycle arrest and apoptotic cell death, which can be triggered by either the extrinsic or intrinsic pathway. Extrinsic apoptosis is characterized by the upregulation of Fas, a death receptor, whose levels were significantly increased at concentrations of 20 and 40 µmol/L after exposure to ISL (6, 12, 24 and 48 h), even after incubation with ZB4, a Fas/FasL system inhibitor. There was also upregulation of membrane-bound Fas ligand (mFasL) and soluble Fas ligand (sFasL). Apoptosis activation was confirmed by cell cycle analysis. The increased fraction of cells in the G0/G1 phase was accompanied by significant massive DNA fragmentation and increased levels of tumor suppressor protein p53 and p21^WAF1^ (an inhibitor of cyclin complexes and cyclin-dependent kinases) in a time- and dose-dependent manner [192,193]. ISL is also a potent inducer of the intrinsic apoptotic pathway through changes in the expression of Bcl-2 family proteins. In A549 cells (at a concentration of 20 µmol/L) and HCC827, H1650 and H1975 cells (at concentrations of 10–40 µmol/L), the ratio of anti-apoptotic protein Bcl-2 to pro-apoptotic proteins Bax and Bim was significantly reduced. Pro-apoptotic Bcl-2 proteins induce the activation of initiator and executioner caspases after mitochondrial damage. This mechanism led to a significant increase in the levels of activated, cleaved caspase-3, with subsequent cleavage of the DNA repair protein PARP, which is considered a marker of apoptosis [194,195].

It is known that NSCLC is associated with various genetic alterations. EGFR mutation and activation of its mediated signaling pathways contribute to the proliferation of tumor cells. It has been shown that ISL is able to inhibit the phosphorylation of this “wild type” or mutated receptor without or after EGF stimulation and directly interact with it. In this study, on multiple lung cancer cells, ISL reduced the phosphorylation and catalytic activity of EGFR downstream kinases such as Akt and Erk1/2 [194]. Akt is part of the PI3K/Akt/mTOR signaling pathway often studied in connection with tumor cells and proliferation. Inhibition of mTOR by repression of PI3K/Akt signaling through incubation with ISL led to suppression of p70 and cyclin D expression, proteins necessary for proliferation and cell cycle progression in A549 cells [195]. Similarly, another transcriptionally active protein is β-catenin, whose mutations and overexpression are associated with various types of cancer, including lung cancer [196]. Suppression of the Wnt/β-catenin pathway by inhibiting the nuclear translocation of β-catenin in tumor cells is one of the mechanisms of the antiproliferative effect of ISL and other active molecules contained in licorice root [197].

Other mechanisms of ISL’s anticancer effect include affecting the tumor microenvironment, invasive capacity, and metastasis of tumor cells. These processes are mediated by proteins that ensure development and wound healing as well as cancer progression. Increased expression of E-cadherin (epithelial marker) and decreased levels of N-cadherin and vimentin (mesenchymal markers) suggest that ISL is capable of inhibiting EMT in A549 cells [195]. Chen et al. [198] identified ISL as the most active migration-inhibitory compound among the studied licorice components, which inhibited the migration ability of H1299, H1975, and A549 cells at concentrations of 3 and 10 µmol/L. ISL also suppressed cytoskeleton reorganization and the number and size of serum-stimulated focal adhesions, which are necessary for cell migration. This inhibitory effect of ISL was probably due to the reduction of FAK (focal adhesion kinase) and cortactin phosphorylation, proteins associated with the cytoskeleton. These proteins are directly phosphorylated by Src, a non-receptor tyrosine kinase associated with cell proliferation and survival, angiogenesis, and invasiveness, and activated in many cancers [199]. The authors found that although ISL was not a direct inhibitor, its metabolite 2, 4, 2′, 4′-tetrahydroxychalcone (THC), which was a product of ISL-treated cells, directly interfered with the Src signaling pathway and inhibited Src kinase activity.

The anticancer effects of ISL have been demonstrated by several studies in vivo. ISL significantly inhibits tumor growth from NCI-H1975 cells in nude xenograft mice [195]. The authors mentioned above, Chen et al., studied the ability of ISL to interfere with Src signaling in vivo. One week after the application of H1299 cells to athymic mice, ISL and its natural metabolite THC were administered. After 3 weeks of treatment, both of these chalcones reduced tumor size (by 60%) and the number and size of metastatic lung lesions. The microscopic analysis showed that tumor cells of mice treated with the vehicle were poorly differentiated and exhibited pathological morphology. Tumor cells in THC and ISL-treated animals were more differentiated, with uniform nuclei, and levels of phosphorylated FAK and cortactin proteins were significantly reduced. These results confirm that ISL and THC inhibit the Src-associated signaling pathway [198].

### 4.4. Cardamonin

Although the positive effects of CAR and plants containing it have been known for years, its antiproliferative activity against lung tumors has only recently been studied in in vitro experiments. Experiments on five lung cell lines confirmed that CAR significantly reduced proliferation in concentrations of 20–40 μmol/L in a time-dependent manner and suppressed the growth of A549 and H460 cell colonies. Analysis of the cell cycle showed a significant increase in the number of cells in the G2/M population along with downregulation of cyclin D1 and Cdk4. Fragmentation of DNA and nuclear condensation after CAR treatment may result in cell cycle arrest and/or cell death. Apoptotic cell death was confirmed by double Annexin V/PI staining and protein analysis (increased levels of the proapoptotic protein Bax and decreased levels of the antiapoptotic protein Bcl-2). In addition, CAR, as well as its two natural analogs, induced apoptosis associated with activation of caspase-3 and PARP cleavage [200,201].

CAR has also been intensively studied in relation to EMT and the tumor microenvironment (TME). In several tumor cell lines A549, H460, LLC, CAR has been shown to effectively reduce the migration and invasiveness of NSCLC cells, even compared to the well-known antitumor drug and immunosuppressant rapamycin [201,202]. In the process of tumor formation, metastasis, tumor cell motility, and drug resistance, EMT is involved, during which there is a disruption of intercellular junctions. This phenotypic change can be observed through changes in the levels of EMT markers, including cadherins and various transcription factors [203]. CAR is capable of inhibiting EMT by upregulating E-cadherin and concomitantly reducing N-cadherin. Similarly, the expression of transcription factors ZEB1 and Snail, which promote EMT [204], was also reduced. Its antimigratory and antimetastatic activity was significant even against TGF-β1-induced EMT, in which N-cadherin was expressed through JNK activation. Phosphorylated active JNK is suppressed by its natural inhibitor PP2A. Downregulation of JNK phosphorylation and upregulation of PP2A indicate dose-dependent CAR-reduced EMT [201,205]. These effects can be mediated by several signaling pathways, of which the Akt/mTOR pathway is often studied. The expression of E-cadherin is partially regulated by mTOR. After treatment of NSCLC cells, the phosphorylation of mTOR upstream kinase Akt, mTOR, and its downstream target ribosomal S6 kinase 1, whose activation dictates various physiological and pathological processes such as growth, cell proliferation, homeostasis, survival, and metastasis, was significantly reduced [206]. Suppression of this pathway led to upregulation of E-cadherin and decreased levels of Snail [201,202,207].

The anticancer effects of CAR were also studied in vivo. Significant reduction in volume and size of tumors induced by *s.c.* or *intraperitoneal* injection of LLC or H460 cells was observed already at a dose of 3.5 mg/kg (reduction of 31.6%). At a dose of 10 mg/kg, tumor growth was inhibited by up to 84.3%, and compared to rapamycin, it also effectively reduced the number of metastatic nodules on the lung surface. Immunohistochemical staining confirmed a reduction in Akt and mTOR phosphorylation, similar to in vitro experiments, and a decrease in Ki-67 positive cells [201,202]. Natural analogs of CAR, such as 4,4′-dihydroxychalcone (DHC) and 4,4′-dihydroxy-2′-methoxychalcone (DHMC), also had significant anticancer effects at a dose of 2 mg/mouse. The authors suggest that their anticancer activity was associated with the inhibition of the NF-κB signaling pathway, which is often activated in lung cancer cells [200].

### 4.5. Licochalcones

One of the criteria for identifying new potential molecules suitable for anti-tumor therapy is selectivity towards tumor tissue. LCA has been shown to be selective and effective in suppressing the proliferation of NSCLC A549 and H460 cells. At a concentration of 40 µmol/L, cell viability decreased by 45–80% after 24 and 48 h of treatment, while LCA demonstrated only minor or insignificant toxicity to normal human lung epithelial cells BEAS-2B [208]. Similar selectivity was observed in other studies on A549 and H1299 cancer cells and healthy normal human embryonic lung fibroblasts [209]. Furthermore, it has been reported that LCA is a potent inhibitor of the cell cycle in the G1 and G2/M phases [208,209,210]. Inhibition of the cell cycle in the G2/M phase is associated with alterations in the expression of proteins regulating the cell cycle. LCA not only downregulated cyclins B1 and D1, cdc2/CDK1, CDK2, CDK4 and cdc25C, essential for the passage of cells through this phase, but also reduced the expression of the ubiquitin ligase MDM2, a major negative regulator of p53, which is considered a proto-oncogene [208,210]. The result of irreparable DNA damage and cell cycle arrest is programmed cell death, most commonly apoptosis and/or autophagic cell death. LCA induces mitochondrial and ER-mediated apoptotic pathways. The intrinsic mitochondrial pathway is mediated by changes in the regulation of proapoptotic (Bax) and antiapoptotic (reduced expression of Bcl-2, Bcl-xL) proteins from the Bcl-2 family in various lung cancer cells (A549, H460) similar to other chalcones. After damage to the mitochondrial membrane, proapoptotic molecules are released from the mitochondria into the cytosol. One of the most important is cytochrome *c*, whose levels significantly increased in the cytosolic fraction after exposure to LCA [208,211,212]. After the initialization of the machinery leading to cell death, there is a cleavage, i.e., activation of caspases-3 and -7, as well as PARP cleavage, which is necessary for cell survival [208,209,213]. Apoptosis of A549, NCI-H1299, and H292 cells induced by ER stress was associated with increased expression and phosphorylation of ER sensors ATF4 and PERK and its downstream proteins EIF2 α and CHOP (C/EBP homologous protein) and also the protein BiP, responsible for proper protein folding. It has been shown that CHOP is an essential molecule for LCA-induced cell death. It is assumed that the induction of ER stress and subsequent autophagy (upregulation of ATG1, ATG3, ATG6, ATG16, and LC3-I/II) and apoptosis after LCA treatment in lung cancer cells was also associated with the upregulation of miR-144-3p. Its knockdown led to the impairment of apoptosis and the antiproliferative effect of LCA. Increased levels of miR-144-3p led to the downregulation of Nrf2 and its downstream γ-glutamylcysteine synthetase catalytic subunit (γ-GCSc) [209,214]. Similarly, Qiu et al. [208] reported the involvement of ER stress in the induction of apoptosis in H460 and A549 cells. After LCA treatment, ATF4 and p-EIF2α proteins were upregulated in a time- and concentration-dependent manner (20–60 µmol/L). In addition to its direct pro-apoptotic effect on cancer cells, LCA is a potent inhibitor of cell death inhibitors such as IAPs and c-FLIPL [213,215]. In A549 and H460 cells, there was a concentration-dependent (10–40 µmol/L) reduction in the levels of c-IAP1 and c-IAP2, which ubiquitinate RIP1 interacting with FADD and caspase-8, and therefore are direct inhibitors of caspase activation [216]. Reduced levels were also observed for the protein XIAP, a direct inhibitor of caspases-3, -7, and -9 and survivin, which is able to inhibit apoptosis by both caspase-dependent and caspase-independent pathways [217]. Another protein whose expression was significantly reduced was c-FLIP_L_, a major anti-apoptotic protein responsible for resistance to cytokine- and chemotherapy-induced apoptosis [218].

Moreover, LCA influences activity of several signaling pathways including PI3K/Akt/mTOR signaling pathway [210], phosphorylation and expression of JNK1, increases phosphorylation of stress kinase p38, activates Erk1/2-dependent protective autophagy [215] or reduces the phosphorylated form of Akt and its downstream transcription factor Sp1 [219], all of which mediate signal transduction from the external to the internal environment of the cell or act through other mechanisms on the growth, proliferation, angiogenesis, metastasis, or resistance of lung tumors. It also inhibits Wnt/β-catenin signaling, as shown on SK-LU-1 lung adenocarcinoma cells, where LCA was the most effective compound in inhibiting the translocation and localization of β-catenin in the nucleus, which can prevent the transcription of target genes [197]. Recently, Gao et al. [213] documented the ability of LCA to modulate also EGFR signaling in NSCLC cell lines. Similarly, to the dysfunction of EGFR signaling, amplification of the Met proto-oncogene (MET) and overexpression of the tyrosine protein kinase Met (c-Met) also play a significant role in tumor development and treatment resistance [220]. In addition to inducing apoptosis in HCC827-GR and PC-9-GR cells, LCA inhibited hepatocyte growth factor (HGF)-induced and uninduced c-Met phosphorylation, as well as Erk1/2 and Akt phosphorylation. By using the proteasome inhibitor MG132, it was found that in addition to reducing c-Met mRNA levels, LCA can also induce c-Met ubiquitination. This effect is probably dependent on the E3 ligase c-Cbl, which is necessary for LCA-induced c-Met degradation [211].

As mentioned above, inhibition of PD-L1 and PD-1-mediated T lymphocyte death is an important mechanism of the anti-tumor activity of many substances that is currently being extensively studied [221]. LCA also effectively inhibited IFN-γ-induced PD-L1 protein expression in A549, NCI-H1299, and NCI-H1650 cells in a concentration-dependent (10 µmol/L) and time-dependent manner (0.5–24 h). Compared to ruxolitinib, a JAK kinase inhibitor, LCA inhibited the translation of PD-L1, probably due to the inhibition of 4EBP1 phosphorylation and activation of the PERK-eIF2a pathway. Since inhibition of 4EBP1 phosphorylation and activation of the PERK-eIF2a pathway were suppressed after NAC administration, with simultaneous suppression of ROS generation, the authors assume that LCA’s antiproliferative effects are also mediated by oxidative stress [222]. In addition to potential molecules in immunotherapy, substances that modulate the tumor microenvironment, invasiveness, and migration ability of cells are also being studied [223]. Although LCA did not show significant toxicity at concentrations of 10 and 20 µmol/L after 24 h exposure to A549 and H460 cells, it significantly inhibited the migration and invasiveness of these cells and downregulated the expression of MMP-1 and MMP-3 at the mRNA and protein levels.

Additionally, LCA has been shown to be an excellent antitumor agent in animal models. Tumor cells such as H226, HCC827-GR, H3255, H1975, HCC827, and A549 cells were implanted in mice and LCA was administered at doses of 10–20 mg/kg for 18 to 24 days. All results showed significant reduction in tumor volume and size. Protein analysis confirmed LCA-mediated inhibition of signaling pathways such as PI3K/Akt, and immunohistochemical staining showed significant reduction in Ki-67 positive cells and decreased expression of c-Met, phospho-EGFR, and survivin in the collected samples [210,211,213].

Similar to LCA, both LCB and LCD have been studied for their potential effect on lung cancer cells. In the treatment of lung cancer, one of the molecular targets is EGFR/ERBB1, belonging to the RTK (receptor tyrosine kinases) family, and Met/HGF, also known as hepatocyte growth factor. Both are often overexpressed in lung cancer patients [224]. Both studies of the authors Oh et al. [225,226] confirmed that these compounds directly interacted with EGFR and Met in gefitinib-sensitive (HCC827) and gefitinib-resistant (HCC827GR) NSCLC cell line models, and also suppressed their kinase activity at a concentration of 5 µmol/L. Molecular modelling and simulation suggest that both bind to the ATP binding region of both proteins. These assumptions correlate with the results of the ATP-P-competitive binding assay, which confirmed that these substances compete for one binding site. LCB significantly suppressed proliferation and viability of both lines at a concentration of 15 µmol/L to 37% and 62% after 48 h, and the IC_50_ of LCD was 17.9 ± 0.97 μmol/L and 19.1 ± 0.5 μmol/L. Proliferation suppression was also confirmed by inhibiting colony formation and cell cycle arrest with an increase in the cell population in the G2/M phase. Cell cycle arrest was associated with the downregulation of cyclin B1 and cdc2 protein kinase and upregulation of the cell cycle progression inhibitor p27 (LCD also upregulated p21). The antiproliferative activity of both chalcones was further clarified by double staining using a mixture of AnnexinV/7-AAD (7-aminoactinomycin D), which showed a concentration-dependent induction of apoptosis by both substances in both types of tumor lines. Increased production of ROS and its inhibition by the antioxidant NAC suggested the initiation of ROS-dependent mitochondrial apoptotic pathways. The increase in the fraction of cells with reduced mitochondrial membrane potential was accompanied by cytochrome *c* release into the cytosol and increased expression of Apaf-1 (apoptotic protease activating factor 1), which are together essential for the formation of the apoptosome and activation of the caspase cascade. Chalcone-induced dysregulation of Bcl-2 family proteins such as Bid, Bcl-xL, Mcl-1, Bad, and Bax significantly contributed to these processes, leading to caspase activation and PARP cleavage. Both chalcones also significantly inhibited the phosphorylation of EGFR, MET, ERBB2, and Akt in both HCC827 and HCC827GR cell lines in a dose-dependent manner. The study from 2019 also suggested the involvement of LCB in the induction of apoptosis associated with ER damage and ER stress (significant increase in levels of CHOP and GRP78/BiP proteins) and the activation of the extrinsic apoptotic pathway, which is characterized by upregulation of death receptors DR4 and DR5 [225,226].

### 4.6. Flavokawain B

Flavokawains are known for many biological effects, including anti-tumor activity [70]. Concerning lung cancer, FKB stopped the growth and proliferation of H460 cells with an IC_50_ of 18.2 µmol/L. Treatment with FKB resulted in cell cycle arrest in the G2/M phase or apoptosis, with typical morphological changes such as rounding and loss of adhesion. At the protein level, there were changes in the levels of members of the Bcl-2 family, upregulation of proapoptotic Bax and downregulation of antiapoptotic Bcl-xL, leading to the release of cytochrome *c* from the mitochondrial space into the cytosol of cells, followed by activation of caspases-3, -7, and -9 and cleavage of PARP. This effect was also accompanied by a reduction in the expression of survivin and X-linked inhibitor of apoptosis (XIAP). Cell death induced by FKB is likely associated with oxidative stress caused by ROS production. Autophagic cell death, mediated by proteins from the Atg (autophagy-related proteins) family such as LC3-I/II, ATG4B, and ATG7, also contributes to the antiproliferative effect of FKB. Autophagy is negatively regulated by the PI3K/Akt/mTOR signaling pathway, which was suppressed in a dose-dependent manner after FKB treatment (2.5–10 µg/mL). FKB is also able to modulate MAPK signaling pathways, of which stress kinases such as JNK or p38 are important [227,228]. These results were also confirmed by Li et al. in paclitaxel-sensitive and resistant A549 cells. Inhibition of Akt phosphorylation led to suppression of P-gp expression, an efflux pump responsible primarily for tumor cell resistance to chemotherapy, which is associated with poor prognosis [229]. Flavokawains are promising molecules not only for anti-tumor therapy but also for chemoprevention with a satisfactory safety profile [230,231].

### 4.7. Garcinol

Garcinol is an extremely potent compound against lung cancer cells. Viability of H460 and H1299 cells exposed to garcinol significantly decreased after 24 h of exposure at a concentration of 2.5 µmol/L. Changes in the distribution of the cell cycle (cell cycle arrest in G1 and S phases) can be explained by the increase in p21 and p27 levels, with subsequent dysregulation of cyclins D1, D3, and E and cyclin-dependent kinases CDK2, CDK4, and CDK6, and suppression of p38 MAPK signaling [232]. In A549 cells treated with garcinol, upregulation of the death receptor DR5 and a decrease in the level of c-FLIP (Cellular FLICE (FADD-like IL-1β-converting enzyme)-inhibitory protein), significant anti-apoptotic protein that inhibits its function by binding to TRAIL receptor 5 (DR5), was observed [218,233].

In a study using H441 and A549 cells to create spheroids that exhibited the lung cancer stem cell (LCSC) phenotype, garcinol reduced the viability of spheroids as well as individual cells with only marginal effects on the survival of non-tumor human lung (bronchial) epithelial BEAS-2B cells. The ability of garcinol to negatively regulate the LCSC phenotype, self-renewal, and multidrug resistance was demonstrated by a reduction in mRNA expression levels of OCT4, BMI1, SOX2, NANOG, NOTCH1, ABCG2, and c-MYC. At the protein level, garcinol reduced the phosphorylation of STAT3 and levels of cyclin D1 and survivin, as well as IL-6-induced phosphorylation of proteins JAK1, JAK2, STAT3, and MAPK. Garcinol also downregulated mediators of the Wnt/β-catenin signaling pathway, including LRP6 (and its phosphorylated form), β-catenin, Dvl2, Axin2 [234]. A similar effect of garcinol on the cellular phenotype in lung cancer cells was recorded through the reduction of ALDH1A1 (Aldehyde Dehydrogenase 1 Family Member A1) expression, a cancer stem-like cell biomarker [235]. This effect was probably mediated by a reduction in C/EBPβ’s ability to bind to the endogenous ALDH1A1 promoter, on which ALDH1A1 expression depends. The experiment also showed garcinol-induced upregulation of DNA damage-inducible transcript 3 (DDIT3), which as a stress responder negatively regulates C/EBPβ [236].

In addition to its individual effect on cancer cells, some studies have shown that garcinol is a suitable compound for combination therapy due to its ability to sensitize cells resistant to erlotinib and cisplatin and reduce TGF-β1-induced EMT (increased expression of E-cadherin and downregulation of N-cadherin and vimentin), probably by affecting the expression of EMT-modulating miRNAs [237]. Garcinol also inhibited the ability of non-homologous end joining (NHEJ) to repair DNA double-strand breaks induced by ionizing radiation, sensitizing A549 cells and increasing the effectiveness of radiation therapy [238].

The antiproliferative activity of garcinol against lung cancer cells has also been demonstrated in animal models. A dose of 5 mg/kg applied for 15 weeks or 15 mg/kg for 40 days significantly reduced the size and weight of mouse tumors, without affecting their body weight, and reduced the number of Ki-67 positive cells. Changes in the expression of DDIT3 and ALDH1A1 were also recorded [234,236]. The antiproliferative and anticancer effects of isoliquiritigenin, flavokawains and garcinol are summarized in Appendix A. (see Appendix A).

### 4.8. Other Chalcones

One of the less studied chalcones is helichrysetin, a chalcone naturally occurring in plants of the genus *Alpinia* and the *Helichrysum odoratissimum* flower. Like other chalcones, helichrysetin also has antiproliferative effects against several tumor cell lines [239,240]. In A549 cells, it induced mitochondria-mediated apoptosis (reducing mitochondrial membrane potential of cells) associated with DNA fragmentation and typical morphological changes and cell cycle arrest in the S phase [241]. Panduratin A, originally isolated from the rhizomes of the plant *Boesenbergia rotunda* (*Zingiberaceae*), used in traditional medicine, has been shown to be a potent compound with antiproliferative activity. At a dose of 5 µg/mL, it induced cell accumulation in the G2/M phase with reduced BrdU incorporation into DNA and increased phosphorylation of histone H3. This often leads to cell death induction, which was accompanied by increased cell membrane permeability, decreased mitochondrial membrane potential, and cytochrome *c* localization shift from mitochondria to cytosol [242]. Further study showed suppression of NF-κB activation in panduratin A-treated A549 associated with the upregulation of p21 and p53 and activation of caspases-3 resulting in PARP cleavage and apoptosis [243]. The ability of panduratin A to modulate NF-κB transcriptional activity has been reported also by other authors. They found that this chalcone inhibits NF-κB translocation into the nucleus as well as TNF-α-induced degradation and inhibitory phosphorylation of IκB, a negative regulator of NF-κB. In addition, inflammatory reaction proteins, cell adhesion molecules, ICAM-1 (intercellular adhesion molecule-1), and VCAM-1 (vascular cell adhesion molecule-1) were downregulated. Its anti-invasive and antimetastatic potential was observed through MMP-2 downregulation [242,243,244]. In addition, several other natural chalcones have undergone investigation as potential drugs for lung cancers. The antiproliferative effect of the chalcone lonchocarpin, isolated from the plant used in traditional medicine *Pongamia pinnata* (L.), was studied in ten lung cancer cell lines. Specifically, apoptosis was induced in H292 cells at a concentration of 20 µmol/L. Changes in the levels of Bax proteins (upregulation) and Bcl-2 (downregulation) led to the release of cytochrome *c* from mitochondria and activation of the apoptotic machinery mediated by caspases-3 and -9. Its anti-tumor potential is also supported by in vivo experiments and 3D-quantitative structure–activity relationship study [245]. A similar mechanism of antiproliferative effect in lung cancer cells, i.e., changes in Bcl-2/Bax ratio and activation of caspase-3 and -9, has also been reported for phloretin and HSYA. Moreover, both chalcones inhibit tumor cell invasion and migration, accompanied by downregulation of MMP-2 and MMP-9. Phloretin and HSYA-induced apoptosis also involve modulation of different signaling pathways such Erk1/2, JNK, p38 MAPK and PI3K/Akt/mTOR. In addition, phloretin improved the antitumor effect of cisplatin and significantly reduced tumor size in A549 lung tumor xenografts at a dose of 20 mg/kg administered every two days for three weeks. On the other hand, HSYA suppressed lipopolysaccharide (LPS)-stimulated EMT and production of proinflammatory cytokines such as TNF-α, IL-6, and IL-1β in A549 and H1229 cells [246,247,248]. Recently, Oh et al. [249] reported that echinatin stimulated ROS production and induced ER stress in both gefitinib-sensitive or -resistant NSCLC cells, and these effects were associated with the G2/M cell cycle arrest and apoptosis induction. Moreover, it suppressed phosphorylation of EGFR and MET as well as their downstream molecules including AKT and ERK. On the other hand, calyxin Y, less studied chalcone isolated from *Alpinia katsumadai*, caused which induced caspase-dependent apoptosis with simultaneous activation of cytoprotective autophagy in NCI-H460 cells. At concentrations from 3.75 to 15 µmol/L, it caused formation of acidic vesicular organelles, upregulation of the Beclin-1 protein, LC-3I/II, and Atg proteins (Atg5, Atg7, Atg5-Atg12) indicated the induction of autophagy. However, co-administration of the autophagy inhibitor 3-MA facilitated apoptotic cell death indicating protective role of autophagy against calyxin Y-induced apoptosis [250].

## 5. Chalcones and Prostate Cancer

Among male cancer types, prostate cancer (PC) represents a second commonly occurred malignance worldwide. In 2020-based GLOBOCAN datasets, there were identified more as 1.4 million new PC cases (7.3%) with mortality about 0.37 million (3.8%) [2]. It is described that the well-established risk factors for PC are family history, hereditary syndromes, and race/ethnicity followed by cigarette smoking, obesity and metabolic syndrome as reviewed [251]. PC progression is known to be androgen dependent and therefore an effective therapeutic strategy used in clinical practice is androgen deprivation therapy (ADT). Despite castrate testosterone levels, PC progression can reach a castration-resistant state, at which point patients are more likely to die of prostate cancer than of other causes. Besides ADT, for metastatic castration-resistant PC, treatment strategies have been developed including the addition of secondary hormonal manipulation (antiandrogens such as bicalutamide, nilutamide, ketoconazole or cortico-steroids) and chemotherapy (mitoxantrone, docetaxel, cabazitaxel, abiraterone acetate, enzalutamide, etc.) as summarized by Teo et al. [252]. At present, several studies are in progress showing some benefits and also failures in areas of systemic therapies and therapeutic combinations with local treatments for high-risk localized PC [253].

In general, an improvement of therapeutic approaches is needed especially for castration-resistant patients due to still short progression-free survival (few months), relapses (in 2-3 years) and bone metastases (80% occurrence in advanced PC). Several in vitro and in vivo studies and some clinical trials have been conducted or are ongoing based on natural products effective in modulation of PC development and progression [254,255].

### 5.1. Xanthohumol

In PC research, prenylated chalcones XH and desmethylXH isolated from *Humulus lupulus* L. have been studied with other presented flavonones for their anti-proliferative properties [256]. Both chalcones showed inhibition potential on PC-3 and DU145 prostate cancer cell lines with relatively low IC_50_ concentrations between 10–50 µM. The most potent chalcone was XH, which showed the lowest IC_50_ values regardless of the type of tested cell line, with different expression levels of estrogen receptors ER-α and ER-β. Subsequent study demonstrated that XH-mediated time-dependent reduction in PC cell viability was accompanied by cell death. However, autophagy rather than a caspase-dependent form was suggested based on morphology observation (formation of vacuoles) and zVAD-fmk assays [257]. In addition, Deeb et al. [258] demonstrated that XH-induced apoptosis in hormone-refractory (AR^+^) PC cell line C4-2 derived from hormone-sensitive cell line (LNCaP) and hormone refractory PC-3 (AR^–^) human PC cell line. The apoptosis was induced through the activation of procaspases-3, -8 -9, PARP cleavage and mitochondrial depolarization with cytochrome *c* release to cytosol. Moreover, XH induced inhibition of phospho-Akt, phospho-mTOR and NF-κB, which indicated involvement of these pro-survival pathways in XH-mediated apoptosis. In addition, the intrinsic mitochondrial pathway was activated based on total Bcl-2 and survivin level reduction after XH treatment. In benign prostate hyperplasia epithelial cells (BPH-1), XH and its oxidized form xanthoaurenol induced apoptosis and cell cycle arrest in S-phase. In this model XH and xanthoaurenol inhibited NF-κB activity by blocking the binding of NF-κB to consensus DNA elements in nucleus [259]. Moreover, XH can act as sensitizer of PC cancer cells (LNCaP) to TRAIL-mediated apoptosis through modulation of extrinsic and intrinsic apoptotic pathways (caspase-3,-8,-9; Bid, Bax, Bcl-xL) [260]. Gieroba et al. [261], using Fourier-transform infrared spectroscopy, also demonstrated that XH caused morphology changes and affected cellular biomolecules leading to apoptosis. One of the potential mechanisms by which XH induces apoptosis is also oxidative stress. It was demonstrated that XH induced superoxide formation, causing mitochondrial damage, cytochrome *c* release and apoptosis [262]. Moreover, XH and its metabolites (isoxanthohumol, 8-prenylnaringenin, and 6-prenylnaringenin) have recently been analyzed by molecular docking to elucidate new potential anticancer targets. Besides known anticancer molecules, a new possible target, the acyl-protein thioesterase 2, was identified [263]. In vivo study showed that XH in low dosage was able to reduce the progression and growth of advanced tumors in the TRAMP mice model. This result is in agreement with in vitro experiments showing inhibition of cell cycle progression, migration, invasion and downregulation of FAK/Akt/ NF-κB pathway in PC-3 model [264].

### 5.2. Butein

In the past decade, the potential of butein in PC research was evaluated. The data showed that butein inhibited the viability and proliferation of prostate PC-3, DU145, LNCaP and CWR22Rm1 cells [265,266]. Moreover, butein treatment caused alteration of cell cycle-associated proteins, e.g., decreased protein expression of cyclins D1, D2, E and cdk2, 4, 6 and an increase in WAF1/p21 and KIP1/27 connected to cell death. Butein-mediated apoptosis was linked with caspase-8, -9, -3 activation, PARP cleavage, an increase in Bax and downregulation of Bcl-2. In addition, butein inhibited PI3K and phosphorylation of Akt at Ser473 and Thr308 in a dose-dependent manner as a part of cell proliferation signaling pathway. IκB and NF-κB inhibition after butein treatment were also described [265,266]. The effect of butein on angiogenesis, migration and metastasis of PC showed that butein treatment suppressed mRNA and protein levels of VEGF and MMP-9 that led to low level of penetration through a Matrigel-coated membrane and inhibition of vascular formation in Matrigel plugs assay [266].

In vivo data also showed that butein inhibited tumor growth in athymic nude mice implanted with AR-positive CWR22Rm1 human PC cells. Decreased levels of cell-proliferation marker (Ki-67) and markers of angiogenesis (VEGF and CD31) were also present in tumor tissue [265].

### 5.3. Isoliquiritigenin

The data showed that ISL reduced growth and induced G2/M resp. G1 cell cycle arrest of PC-3 and LNCaP PC cells [267,268] or DU145 cells [269]. Moreover, the analyses of G2/M phase regulatory proteins status and activation after ISL treatment showed increased p-CDC2, but no changes in total CDC2 protein levels and an increase in cyclin B1 with downregulation of CDC25C in a concentration-dependent manner in DU145 cells. In C4-2 PC cells, ISL activated p38 MAPK and ERK signaling pathways and did not affect AMPK activation that regulates cell growth, proliferation, differentiation and survival in response to extracellular stimuli [270]. Furthermore, Kwon et al. [271] showed that ISL inhibited basal and EGF-induced cell migration, invasion and adhesion of DU145 cells. It was shown that ISL suppressed EGF-stimulated production of MMP-9, uPA, TIMP-1 and VEGF as pro-migratory factors. Moreover, ISL inhibited cell adhesion by downregulation of integrin-α2, ICAM and VCAM. The decreased EGF-induced phosphorylation of Akt and JNK was also described after ISL treatment supporting anti-migratory potential.

In addition, the in vivo results showed therapeutic efficacy of ISL. The tumor growth, volume and size were significantly reduced after ISL application of PC-3 xenograft model in male nude mice. ISL treatment also induced cell apoptosis in vivo linked with caspase-3 activation and downregulation of CDK1 and cyclin B1.

### 5.4. Cardamonin

Another chalcone that possesses anti-cancer activity against PC is CAR. The first evidence revealed that CAR expressed cytotoxic activity against PC-3 prostate cancer cells. Pascoal et al. showed that CAR induced DNA fragmentation associated with cell death and downregulation of NF-κB1 after 12, 24 and 48 h treatment [272]. More mechanisms associated with CAR-mediated cell death of PC androgen-independent (DU145) and androgen-dependent (LNCaP) cell lines were described. Zhang et al. [273] showed that CAR suppressed STAT3 activation through inhibition of upstream JAK2 kinase, inhibition of STAT3 dimerization and DNA binding ability. Moreover, CAR downregulated the expression of angiogenic, proliferative and metastatic gene products (Bcl-xL, Bcl-2, Survivin, XIAP, VEGF, COX2 and MMP-9) in a time-dependent manner. In addition, CAR decreased the expression of cyclin D1, CDK4, cyclin E and CDK2 proteins in a time-dependent manner and induced apoptosis associated with caspase-8, -9, -3 activation and PARP cleavage. It was also proven that CAR inhibited PC cell migration and invasion regardless CXCL12 (chemokine ligand C-X-C motif chemokine ligand 12) stimulation.

### 5.5. Licochalcones

Chalcone isolated from Licorice also possesses anti-cancer activity against PC. The first insight indicated that LCA significantly suppressed PC-3 cell growth rather than induced cytotoxicity in time-dependent manner. Cytostatic effects were associated with the G2/M cell cycle arrest linked with downregulation of cyclin B1/cdc2 and Rb [274]. The latest study on *Glycyrrhiza echinata* roots showed that extract from roots contains eight compounds including four chalcones—ISL, echinatin, LCB and tetrahydroxylmetoxychalcone (THMC). As the results demonstrated, methanol root extract and all chalcones had cytotoxic effects on PC-3 prostate cells. The comparison with normal prostate cells, the THMC showed the most selectivity (selectivity index = 5.195). In addition, analyses showed that both echinatin and THMC exhibited condensed nuclei, significant increases in the percentage of apoptotic cells and an increase in the ratio of G1 and G2/M cell cycle phase. Suppression of the migration was also confirmed [275].

### 5.6. Flavokawains

Three types of flavokawains (FKA, FKB and FKC) were identified from *Piper methysticum* (Kava-kava). FKA and FKB were also evaluated regarding to PC. It has been shown that FKA had inhibitory effects on the growth of PC cell lines DU145, PC3 and 22Rv1, while no effect was observed on normal prostate epithelial cells (PrECs) and prostate stromal cells (PrSCs). Moreover, obtained data revealed that this inhibition was relevant to Rb protein absence (DU145) and was associated with deNEDDylation and Skp2 degradation independent of Cdh1, pRb deletion or p53 mutations. On the other hand, functional cullin1 protein was required. In vivo evaluation in TRAMP transgenic mouse model also showed that FKA effectively inhibited proliferation and induced apoptosis in pre-cancerous and cancerous cells in this model. In addition, distant organ metastases were not observed after FKA treatment [276]. Recently, other mechanisms were reviewed on PC-3 cells. Cell cycle analysis showed that FKA induced G2/M phase arrest associated with tubulin polymerization inhibition and survivin downregulation. Moreover, intracellular levels of glutamine, glutamic acid and proline in PC-3 cells decreased after FKA treatment that suggesting changes in glutamine metabolisms. These changes led to decreased GSH content and increased ROS levels [277]. Most recent research also demonstrated that dietary FKA reduced in vivo growth of xenograft tumors generated from highly tumorigenic CSCs (PC stem cells). FKA inhibited the expression of stem cell markers (Nanog, Oct4, CD44), neddylation of Ubc12 and the expression of c-Myc in both prostaspheres and tumor tissues [278]. Furthermore, Tang et al. [279] demonstrated that FKB predominantly inhibited growth of AR-negative than AR-positive, hormone-sensitive PC cell lines. Moreover, it was shown that FKB-mediated apoptosis was linked with upregulation of Bax expression and upstream activators as Bim, Puma and DR5; and downregulation of survivin and XIAP expression. In vivo data also confirmed an anti-tumor potential of FKB. Moreover, Li et al. [280] showed that FKB combined with proteasome inhibitor bortezomib enhanced pro-apoptotic potential through downregulation of Skp2 expression. In addition, upregulation of p21/WAF1, p27/Kip1 and activation of the caspase-mediated apoptotic pathway were presented.

### 5.7. Garcinol

This polyisoprenylated benzophenone induced cytotoxicity in DU145, PC-3, LNCaP and C4-2B PC cells [281,282]. Garcinol in dose- and time-dependent manner induced apoptosis with caspase-9, -3 activation, PARP cleavage and DNA fragmentation in PC-3 cells. The intrinsic apoptotic pathway activation after garcinol treatment was confirmed by reduced mitochondrial membrane potential, increased Bax and downregulated Bcl-2. Moreover, garcinol was able to induce in minority also autophagy cell death by increased LC3B I/II but did not change the expression of beclin-1, Atg 12, and Atg 5. In addition, garcinol upregulated phosphorylation of mTOR, GSK-3β, PI3K, Akt, PDK1 and downregulated Erk1/2a in time-dependent manner [282]. Ahmad et al. [281] also showed that garcinol inhibited NF-κB activity. In vivo (PC-3 xenograft nude mice), garcinol decreased tumor volume and size and induced apoptosis [282].

### 5.8. Other Chalcones

Phloretin induced morphological changes, inhibited PC cell (LNCaP, CWR22Rv1, PC-3, and DU145) viability, induced G2/M cell cycle arrest and apoptosis linked with decreased protein levels of cyclin B1, XIAP, and Bcl-2, while increased the protein levels of Bax, cleaved caspase-8, -9, -3 and PARP in a dose-dependent manner. Moreover, phloretin downregulated EGFR activation and PI3K/Akt and Erk1/2 pathway, decreased the phosphorylation level of GSK3β and transcriptional activity of Sp 1 with its targeted genes (Sp3/4, XIAP, VEGF, survivin, cyclin D1, and AR) [283]. In addition, phloretin was able to inhibit colony formation and migration of the PC-3 and DU145 PC cells. Kim et al. [284] also described that phloretin significantly increased the level of ROS, and showed that ROS-related antioxidant enzyme gene expression (SOD2, Catalase, Gpx1, Gpx3 and CISD2) decreased. It was also documented that phloretin altered the Wnt/β-catenin pathway by downregulation of β-catenin, TCF4, FoxA2 and Twist (EMT-related proteins). Szliszka et al. [285] demonstrated that phloretin markedly augmented TRAIL-mediated apoptosis in LNCaP cells. In vivo data on xenograft tumor models of PC-3 cells in nude mice showed that dietary phloretin inhibited the growth of tumors and reduced volume in the high-dose and low-dose regimen. Moreover, tumor tissue analyses showed decreased proliferation marker Ki-67, Sp1, Sp3/4, survivin and cycline D1 and increased levels of cleaved caspase-3 and PARP [283].

Naturally occurring chalcone IBC showed cytotoxic potential on PC. It has been documented that IBC inhibited the viability of PC-3 cells, induced apoptosis visualized by cell shrinkage, chromatin compaction, cytoplasm condensation, nuclear fragmentation and phosphatidylserine externalization. Moreover, IBC induced caspase-3 activation and ER stress pathway (upregulation of GRP78, ATF4, XBP-1, and Chop mRNA) associated with TrxR (thioredoxin reductase) inhibition and ROS production [286]. In addition, TRAIL-mediated apoptosis and Akt signal pathway inhibition was also described in PC cells treated with IBC [287,288].

Several chalcones have been isolated and identified from plant family *Zingiberaceae* (*Alpinia zerumbet, Boesenbergia rotunda, Kaempferia pandurate*), including pinostrobin, boesenbergin A and panduratin A, which were tested against PC. Panduratin A from *Kaempferia pandurate* showed significant cytotoxicity on PC-3 and DU145 PC cells. It induced externalization of phosphatidylserine (early apoptotic marker), activated caspase-8, -9, -6, and -3 with subsequent PARP cleavage. Except for caspase-9 activation, activation of intrinsic apoptotic pathway by panduratin A was confirmed by increased Bax and decreased Bcl-2, Bid protein expression. The death receptor pathway was also activated, as it showed increased expression of FADD (FAS-associated death domain) and TRAIL. Moreover, panduratin A induced G2/M cell cycle arrest associated with dose-dependent decrease in the levels of cdc25C, cdc2, cyclin B1/D1/E1 and cdks 2, 4 and 6. In addition, panduratin A increased levels of p27Kip1 and p21WAF1/Cip1 protein, associated with cell cycle arrest and apoptosis in PC cells [289]. Boesenbergin A from *Boesenbergia rotunda* and pinostrobin from *Alpinia zerumbet* were tested on some cancer cell lines including PC and showing significant cytotoxicity [290,291].

## 6. Chalcones and Renal Cancer

Renal cell carcinoma (RCC), also known as renal cell cancer or renal cell adenocarcinoma, is the most common type of kidney cancer [292,293,294]. The prevalence of RCC is rising in most countries globally, and it has a relatively high fatality rate [295,296]. The diagnosis of RCC frequently occurs in its initial phase, wherein the cancer is limited to the kidney and is of a small size. Clear cell RCC accounts for the majority (65–70%) of RCC subtypes, followed by papillary RCC (15–20%) and chromophobe RCC (5–7%), respectively [297].

### 6.1. Isoliquiritigenin

Using ISL-treated Caki RCC cell lines, Kim et al. [298] observed decreased cell viability and induction of apoptosis associated with the activation of caspase-3, -7, and -9, and PARP cleavage. In addition, ISL-induced apoptosis has been associated with upregulated pro-apoptotic protein Bax expression and diminished expression of anti-apoptotic proteins Bcl-2, and Bcl-xl, with subsequent cytochrome c release. Furthermore, the exposition of CAKI cells to ISL also resulted in the suppression of STA3 activity followed by reduced expression of STAT3 downstream molecules such as cyclin D1 and cyclin D2. On the other hand, pretreatment with NAC modulated the inhibitory activity of ISL indicating a role of ROS in the antiproliferative effect of this natural chalcone. In another study, ISL has been found to significantly inhibit the RCC 786-O Cells proliferation, migration, and invasion. Detailed analyses showed that the exposition of 786-O cells to ISL resulted in the induction of autophagy due to inhibition of The PI3K/Akt/mTOR signaling pathway [299]. In an animal study, ISL significantly reduced the number of metastatic nodules in the lungs in mice bearing mouse RCC Renca cell line. In vitro studies showed that ISL reduced Renca cell viability and increased the activity of immune cells including macrophages and splenic lymphocytes [300].

### 6.2. Broussochalcone A

Lee et al. [292] demonstrated the apoptotic mechanisms of broussochalcone A in A498 and ACHN cells, two types of RCC cell lines. The results of this study showed decreased cell viability and cell growth in broussochalcone A-treated cells. Furthermore, cell cycle arrest at the G2/M phase and induction of apoptosis was observed in cells exposed to broussochalcone A. Further Western blot analyses showed that broussochalcone A either upregulated (cleaved PARP, FOXO3, Bax, p21, p27, p53, phosphorylated p53, active forms of caspase-3, -7, and -9) or downregulated (pAkt, Bcl-2, and Bcl-xL) several proteins indicating activation of mitochondrial pathway of apoptosis. In addition, increased levels of ROS were detected, and could be associated with the activation of the FOXO3 signaling pathway.

### 6.3. Nephroprotective Effect of Chalcones in Cisplatin-Induced Nephrotoxicity

As mentioned previously, the antiproliferative effect of XH has been demonstrated in various types of tumors. Surprisingly, there is no relevant study focused on the antiproliferative effect of XH in RCC. On the other hand, in the field of anticancer therapy, the nephroprotective effect of XH could be interesting. In the study of Li and co-workers [301] the model of cisplatin-induced nephrotoxicity was used. Results of this study showed significant attenuation of kidney damage induced by intraperitoneal administration of cisplatin. It has been documented by decreased levels of plasmatic creatinine, reduced levels of ROS and malondialdehyde as well as by increased levels of GSH and SOD. Furthermore, XH also significantly decreased the levels of inflammatory cytokines including TNF-α, IL-1β, and IL-6 in kidney tissues. Similarly to XH, the nephroprotective effect of panduratin A has also been studied in cisplatin-induced kidney injury in mice. Thongnuanjan et al. [302] investigated the effect of panduratin A on the toxicity of cisplatin in vivo in mice and in vitro in human renal cell cultures using RPTEC/TERT1 cells. The results showed a significant nephroprotective effect of panduratin A. In an animal model, it improved cisplatin-induced renal injury via inhibition of ERK1/2 and caspase-3. In addition, in RPTEC/TERT1 cells panduratin A prevented cisplatin-induced apoptosis and modulated activation of ERK1/2 signaling and caspase-3. The renoprotective effect in cisplatin-induced injury has also been demonstrated in ISL-treated animals. As was reported, ISL decreased levels of creatinine and serum nitrogen in comparison of cisplatin-treated animals. Moreover, levels of pro-inflammatory cytokines such as IL-1β, IL-6 and TNF-α were significantly decreased in ISL-treated animals [303]. Additionally, the preventive effect of ISL against cisplatin-induced renal cells was also documented in animal kidney cells. Preventive effect of ISL was associated with decrease in ROS production, maintaining the GSH levels and GR activity and increased heme oxygenase-1 expression [304,305].

## 7. Chalcones and Bladder Cancer

Bladder cancer (BLCA) is one of the most common malignancies. According GLOBOCAN 2020, BLCA ranks as the tenth the most frequently occurring cancer worldwide [2]. Although great progress has been made in the treatment of bladder cancer, the search for new promising molecules with antitumor activity is still one of the greatest challenges.

### 7.1. Butein

Butein has been found to exhibit anti-proliferative effects against wide range of cancers [306]. However, there is only one study related to BLCA. Zhang et al. [307] studied antimigratory and anti-invasive effect of butein in human bladder cancer cells. Western blot analysis showed decreased expression of phosphorylated ERK1/2 and inhibition of NF-κB signaling pathway. In addition, butein also reverted TNF-α-induced EMT in bladder cancer cells as documented by decreased vimentin expression and increased expression of E-cadherin in butein-treated cells. Furthermore, this chalcone also decreased expression of genes involved in EMT.

### 7.2. Isoliquiritigenin

Antiproliferative and pro-apoptotic effects of ISL have been studied in human bladder cancer T24 cells. Twenty-four-hour treatment of T24 cells resulted in chromatin condensation, nuclear fragmentation, cellular shrinking or loss of cell membrane integrity. Furthermore, ISL upregulated pro-apoptotic protein expression (e.g., Bax, Bim, Apaf-1, caspase-9 and caspase-3) and downregulated expression of the antiapoptotic Bcl-2 protein, decreased mitochondrial membrane potential and increased cyclin dependent kinase 2 activity [308]. Later, Huang et al. [309] reported that ISL caused arrest of cell cycle at G1 phase in T24 cells. It also significantly reduced cell migration and downregulated expression of proliferating cell nuclear antigen, which plays a role in cell transition from G1 phase to S phase. In another study, ISL was reported to induce cell death in 5637 human bladder cancer cell line. Treatment of these cells with ISL led to induction of apoptosis associated with caspase-3 and -9 activation, modulation of expression of Bcl-2 family proteins and increased levels of intracellular ROS [310].

### 7.3. Licochalcones

In the study by Yuan et al. [311], LCA was found to be strong inhibitor of T24 cell proliferation. Moreover, it potently initiated apoptosis, as confirmed by PARP cleavage, caspase-3 and -9 activation and mitochondrial dysfunction. In addition, ER stress response was induced in LCA-treated T24 cells. Furthermore, increase in intracellular ROS and decreased GSH/GSSG ratio indicated possible role of oxidative stress in the antiproliferative effect of LCA. The involvement of oxidative stress in antiproliferative role of LCA also confirmed further study where NAC significantly attenuated LCA-induced inhibition of proliferation in T24 cells [312]. Later, a study by Yang et al. [313] reported possible association between LCA-induced apoptosis and an increase in intracellular Ca^2+^ concentration. It was declared that LCA induced mitochondrial dysfunction, stimulated ROS production and modulated expression of several genes associated with apoptosis including Bax, Bim, Apaf-1, caspase-9 and -3 with a concomitant increase in intracellular Ca^2+^ concentration. Other authors reported the arrest at the G2/M phase of the cell cycle in LCA-treated T24 cells. Molecular analyses showed suppression of cyclin A, cyclin B1, and Wee1 expression. On the other hand, this chalcone increased the expression of cyclin-dependent kinase (Cdk) inhibitor p21WAF1/CIP1. Similarly to previous articles, LCA increased ROS production and caused mitochondrial dysfunction [314]. Just like LCA, LCB has also been referred as an antiproliferative agent against bladder cancer. The study by Yuan et al. [315] documented the effect of LCB in two human bladder cancer lines, T24 and EJ, and murine transitional cell carcinoma cell line MB49. LCB significantly suppressed the growth of human cancer cell lines in a dose-dependent manner. It blocked cell cycle progression at G1 phase and this effect was associated with decreased expression of DK1 and CDK2 as well as Cdc25A and Cdc25B, and other proteins involved in cell cycle regulation. In addition, modulation of Bcl-2 protein expression, downregulation of survivin expression, and PARP cleavage have also been observed. Moreover, in vivo experiments in mice bearing MB49 xenograft showed an anticancer effect of LCB. Tumor size and weight were significantly reduced in LCB-treated animals. Another study has shown that LCB possesses anti-migratory, anti-invasive and anti-adhesive activity [316]. LCB suppressed protein expression and activity of MMP-9, as well as the protein level of NF-κBP65 and nuclear translocation of NF-κB in T24 cells. Later, the antiproliferative and pro-apoptotic effect of LCC has also been reported. In T24 cells LCC inhibited cell proliferation and induced apoptosis in a concentration-dependent manner. It decreased levels of anti-apoptotic members of Bcl-2 family proteins (i.e., Bcl-2, Bcl-w and Bcl-XL) and increased expression of pro-apoptotic members including Bax and Bim on the genetic level. Moreover, some markers of apoptosis, such as PARP-cleavage, activation of caspase-3 or externalization of phosphatidylcholine, have also been observed [317].

### 7.4. Flavokawains

The antiproliferative and anticancer effects of flavokawains have also been studied in bladder cancers. Zi and Simoneau [318] studied the antiproliferative effect of FKA, FKB and FKC isolated from kava extract using RT4, T24 and EJ human bladder cancer cell lines. They found strong growth-suppressive effect in all flavokawains with IC_50_ significantly lower in T24 and EJ cells with mutant p53. Further analyses showed that FKA induced apoptosis mostly via intrinsic pathway as documented by activation of caspase-9 and -3, increase in Bax/Bcl-xL ratio, decrease in mitochondrial membrane potential and release of cytochrome c into cytosol. Later, this research group described different mechanisms of action in human bladder cancer cell lines with wild-type and mutated p53. In a p53 wild-type RT4 cancer cell line, treatment with FKA led to G1 cell cycle arrest due to inhibition of CDK2 activity. On the other hand, six cell lines with mutated p53 FKA caused cell cycle arrest at G2/M phase associated with reduced expression of CDK1-inhibitory kinases with subsequent CDK1 activation [319]. Potential of FKA to suppress bladder tumorigenesis was confirmed using UPII-SV40T transgenic model. FKA (6 g/kg of food) significantly increased survival of male UPII-SV40T transgenic mice. About 38% and 64% of animals fed with control diet and FKA-supplemented diet, respectively, survived beyond 318 days. In addition, immunohistochemical analyses showed decreased number of Ki-67 positive cells and increased level of apoptotic cells in FKA-treated animals. Similar chemoprotective effect of FKA has also been reported in a recent study using UPII-mutant Ha-ras transgenic mouse model. Like in previous studies, FKA feeding significantly reduced risk of the development of papillary urothelial cell carcinoma [320].

As mentioned above, PRMTs regulate numerous crucial cellular processes. Among other things, they are also involved in cancer development, progression, and aggressiveness [321]. Recently, Zhang et al. [322] reported that PRMT5 can play role in bladder cancer proliferation and invasivity and could be a therapeutic target. The study by Liu et al. [323] showed that FKA specifically inhibited PRMT5 and this effect was even stronger when compared with EPZ015666 and GSK3326595, two known inhibitors of PRMT5. In addition, this effect has also been confirmed in vivo.

### 7.5. Other Chalcones

Increased expression of glucose transporter proteins (GLUT) and glycolytic enzymes in neoplastic cells is linked to enhanced glycolysis, which is frequently observed in rapidly growing solid tumors. A natural dihydrochalcone, phloretin has been reported to inhibit glucose transmembrane transport in Fischer bladder cell carcinoma cell lines resulting in suppression of tumor cell growth and a decrease in the size of tumors in experimental animals [324,325]. In a further study, the mechanism of the antiproliferative and apoptosis-inducing effect of chalcone was studied in T24 and HT-1376 human bladder cancer cell lines. In both cell lines, chalcone blocks cell cycle progression at the G2/M phase. This effect has been associated with downregulation of cyclin B1, cyclin A and Cdc2 and upregulation of p21 and p27 proteins. Moreover, chalcone treatment led to overexpression of proapoptotic Bax and Bak proteins and, on the other hand, decreased expression of antiapoptotic Bcl-2 and Bcl-XL proteins with subsequent release of cytochrome *c* and activation of caspase-9 and -3. In addition, this chalcone inhibited both nucleic translocation and activation of NF-κB [326].

## 8. Chalcones and Melanoma

Cutaneous melanoma is one of the most aggressive and lethal cancers arising from melanocytes, and has shown an increasing incidence in the past few years. It most often occurs in Australia and New Zealand, followed by North America and Northern/Western Europe, usually in the Caucasian population [327]. Approximately 100,350 cases of newly diagnosed primary malignant tumors (excluding non-melanoma skin cancer) are skin melanoma, and the incidence is still rising. Although melanoma only accounts for about 5% of skin cancers, it results in more than 75% of deaths [328].

Melanoma primarily involves the skin, and can occur from a pre-existing pigmented mole or can occur also de novo. The skin signs of melanoma are often based on ABCDE criteria (asymmetry, border irregularity, color variety, diameter, evolution) or an “ugly duckling” sign, in that case, the lesion usually looks different from the others. The ABCDE criteria do not apply to all melanomas, and in practice, the AC rule for melanoma (asymmetry, color variation) could be more sensitive. The most important factors are prevention and regular skin investigation [329]. According to the high metastatic potential of melanoma, an early diagnosis with prompt and effective treatment are crucial in terms of disease prognosis. Therapy of melanoma includes surgery, chemotherapy, radiation, targeted therapies using inhibitors directed against signaling components and immunotherapies with monoclonal antibodies.

Similarly to other cancer types, antiproliferative effect of chalcones have also been studied in melanoma skin cancers.

### 8.1. Xanthohumol

In a recent study [330], an antiproliferative and antimetastatic activity of XH has been studied using murine melanoma cell line B16-F10. Results of in vitro experiments showed dose-dependent inhibition of proliferation, colony formation, and migratory activity of XH-treated melanoma cells. Because of the high metastatic potential of melanoma, the effect of XH on hepatic metastasis in a murine model has also been analyzed. In XH-treated animals, a significant decrease in the number and size of hepatic metastases was observed. Furthermore, XH significantly decreased MIA (melanoma inhibitory activity) protein which plays an important role in the progression and metastasis of melanomas. In addition, the immunohistological analysis reported significantly reduced levels of Ki67, a marker of cellular proliferation. In another study, XH-induced cytotoxicity in B16-F10 cells was associated with the induction of ER stress and increased protein ubiquitination. In addition, significant decrease in phosphorylated p38, JNK and ERK proteins was found in XH-treated melanoma cells [331].

Although there are limited data about the bioavailability of XH, it is generally accepted that XH has low bioavailability in humans and animals [332]. The study by Fonseca et al. [333] was focused on the comparison of the antiproliferative effect of XH and XH encapsulated in poly lactic-co-glycolic acid (PLGA) nanoparticles in the B16F10 melanoma cell line. The results showed similar activity of both XH and encapsulated XH on cell viability and migration and demonstrated the suitability of this nano-carrier for XH delivery.

### 8.2. Flavokawain B

In the study of Hseu et al. [334], the antiproliferative and pro-apoptotic effects of FLB were studied in human epithelial melanoma cell line A375 and human skin lymph node-derived melanoma cell line A2058 cells. In both cell lines, FKB reduced cell survival in micromolar doses. In addition, FKB also modulated the expression of several proteins involved in apoptosis and autophagy. FKB activated caspase-3 followed by PARP cleavage in a dose-dependent manner. Moreover, the expression of Bcl-2 was significantly downregulated in both cell lines. These results indicated FKB-induced apoptosis in A375 and A2058 melanoma cells. On the other hand, some results such as LC3-II accumulation, formation of acidic vesicular organelles or inhibition of Akt/mTOR signaling pathway showed possible involvement of autophagy in FKB-induced melanoma cell death. The data of this study indicate that communication exists between apoptosis and autophagy induced by FKB in melanoma cells. Moreover, FKB was also found to inhibit tumor growth in A375 xenografted animals.

### 8.3. Licochalcones

A study by Zhang et al. [335] evaluated the mechanism of antiproliferative action of LCA A in A375 and B16 melanoma cells. In both cell lines, LCA suppressed proliferation in a dose- and time-dependent manner. Furthermore, LCA also inhibited the expression of microphthalmia-associated transcription factor (MITF), which plays important role in melanogenesis. In addition, the activation of caspase-3 in LCA-treated melanoma cells was observed. However, further analyses showed that LCA-induced decrease in cell proliferation is rather due to the activation of autophagy as evidenced by suppression of the mTOR pathway and increased expression of LC3-II, beclin1 and ATG5 proteins. Another study revealed that LCD also possesses pro-apoptotic, anti-migratory and anti-invasive activity in A375 cells [336]. LCD downregulated antiapoptotic Bcl-2 protein while upregulated pro-apoptotic protein Bax and caspase-9 and -3. These effects were associated with a decrease in mitochondrial membrane potential and the generation of ROS. Additionally, the inhibition of migration and invasion has been associated with significant reduction of MMP-2 and MMP-9 expression and activity. Furthermore, LCD decreased tumor progression in vivo in mice bearing melanoma B16F0 cells.

### 8.4. Isoliquiritigenin

The antiproliferative and anticancer effect of another natural chalcone, ISL, has also been investigated. In mouse melanoma cells B16-F10 ISL decreased proliferation via reduced expression of enzymes related to glycolysis such as GLUT 1/4, hexokinase 2, pyruvate kinase M2 and lactate dehydrogenase. Furthermore, a decrease in mitochondrial membrane potential, increased cellular ROS levels, and downregulation of HIF-1α in ISL-treated cells were also observed [337]. In a further study, Chen et al. [338] reported ISL-induced apoptosis and mitochondrial dysfunction associated with the decrease in the mitochondrial protein mitoNEET expression, leading to increased production of ROS. Concomitantly with ROS overproduction, the increased number of apoptotic cells has also been recorded. Later, Xing et al. [339] confirmed the pro-apoptotic effect of ISL in melanoma cells. Moreover, they found that ISL suppressed the migration of melanoma cells and this effect was associated with miRNA27a downregulation. Further analyses also showed that ISL prevented EMT as evidenced by increased expression of E-cadherin and decreased expression of vimentin. It was suggested that these effects can be associated with POU2F3, a tumor suppressor gene that is negatively regulated by miRNA27a. Other studies documented that the antiproliferative effect of ISL can be related to its ability to inhibit some other microRNAs including miR-301b [340] or circ_0002860/miRNA-431-5p [341].

### 8.5. Cardamonin

Berning et al. [327] documented the results of an in vitro study, where the antiproliferative effect of CAR was studied in an A375 cell line and compared to the effect on normal human epidermal melanocytes (NHEM) or normal human dermal fibroblasts (NHDF). The viability and invasivity of the A375 melanoma cells were significantly lowered after 24 h treatment with CAR. On the other hand, non-cancer cells at the CAR concentrations used were minimally affected. The strongest cytotoxic effect was observed with 20 μM of CAR, as only a fraction of about 5% of tumor cells still survived after 24 h of treatment. Later, Yue et al. [342] confirmed the antiproliferative and pro-apoptotic effects of CAR in human melanoma M14 and A375 cells. They demonstrated that CAR induced both the intrinsic and extrinsic apoptosis pathways.

### 8.6. Panduratin A

In the study by Lai and co-workers [343], proteomic analysis showed that panduratin A deregulated 296 proteins in A375 cells. It caused mitochondrial dysfunction, loss of mitochondrial membrane potential and decreased ATP production. Moreover, panduratin A activated UPR (unfolded protein response) signaling and mediated ER stress-induced cell death. Later, this research group, found activation of autophagy in panduratin A-treated A375 cells. However, it seems that autophagy, in this case, played a cytoprotective function, because the co-treatment of panduratin A with an autophagy inhibitor chloroquine significantly increased panduratin A-induced apoptosis [344].

## 9. Conclusions

This review focuses on the potential of natural chalcones as effective anticancer agents against various types of cancer, such as breast, gastrointestinal, prostate, melanoma, lung, renal and bladder cancer. The tumor-suppressive effect of these polyphenols is mediated by the multitargeted abilities of chalcones such as cell cycle arrest and apoptosis induction, modulation of autophagy mechanisms, inhibition of several transcription factors or modulation of signaling pathways associated with cell survival or death (Figure 3). It is noteworthy that numerous in vitro and in vivo experiments have demonstrated the selective anticancer properties of these natural compounds, indicating their potential as targeted cancer-killing agents. However, despite their potential as anticancer agents, natural chalcones have limited bioavailability due to their low solubility. Addressing this issue is essential for researchers to be able to explore its full potential as a pharmaceutical drug candidate and open the way for clinical trials. Finally, due to the continually rising prevalence of malignant diseases worldwide, it is crucial to continue researching and exploring the potential of natural compounds, including chalcones, to develop effective treatments for cancer.

## Figures and Tables

**Figure 1 ijms-24-10354-f001:**
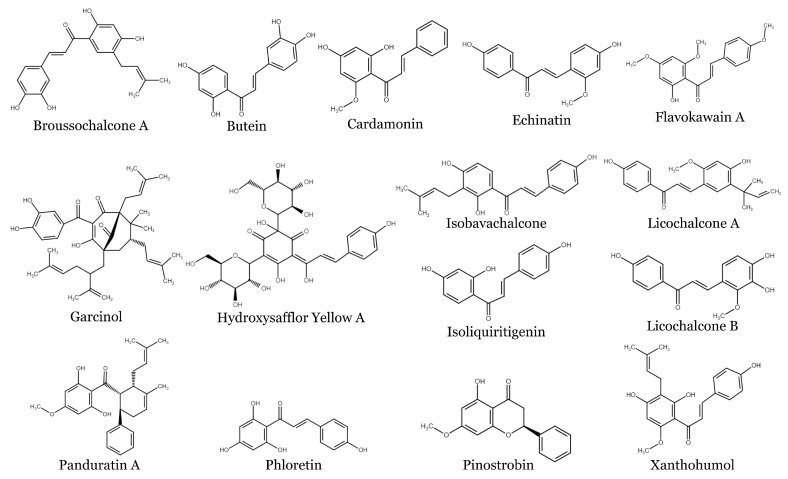
Structures of selected naturally occurring chalcones discussed in this review.

**Figure 2 ijms-24-10354-f002:**
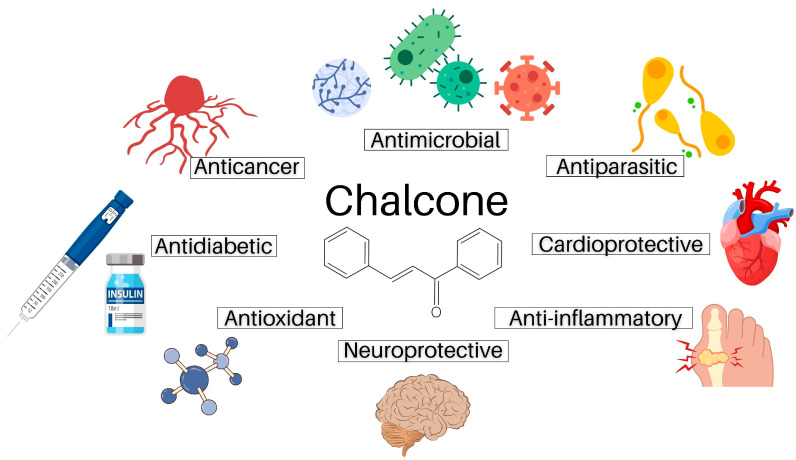
The pharmacological activities of natural chalcones.

**Figure 3 ijms-24-10354-f003:**
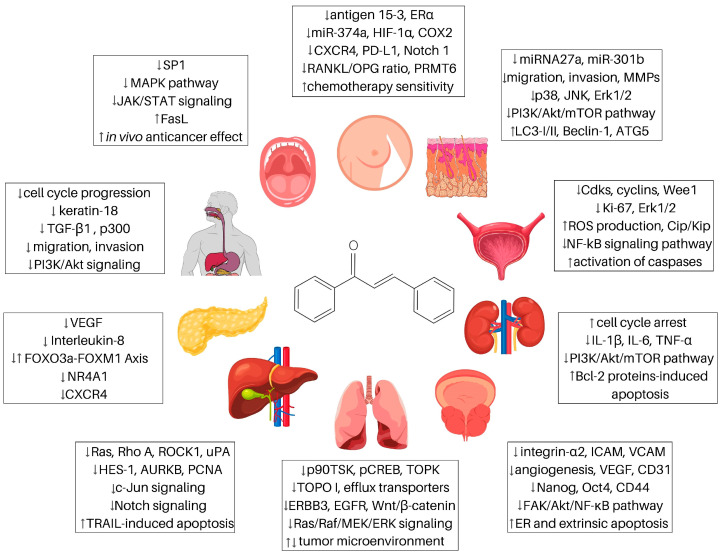
Chalcones and their mode of antiproliferative action. Arrows indicate an increase (↑) or decrease (↓) in the levels/activity of the molecules.

## Data Availability

Not applicable.

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
