# Peer review of "Anticancer Potential of Natural Chalcones: In Vitro and In Vivo Evidence"

_ijms, 2023, doi:10.3390/ijms241210354_

Round 1

Reviewer 1 Report

Chalcones are among the most therapeutically-active flavonoids. Multiple chalcones are currently at the stage of preclinical and clinical studies, and they are employed as antidiabetic, anticancer, anti-inflammatory, antimicrobial, antioxidant, antiparasitic, psychoactive, and neuroprotective drugs. This review addresses the antiproliferative properties of chalcones in a plethora of various cancers, describing their mechanism of action. The review is excellently written, well structured and referenced. I recommend it for the publication after minor corrections:

- Tables 1, 2: there is too much white space in the left column. This causes the tables to be several pages long, and it is difficult to read the tables without printing out. The structure of the tables should be improved.

- Table 3: the same, first 3 columns are almost empty, too much white space. Its structure should be improved.

- The text in Figure 2 should be in a higher resolution than it is now.

- The first sentence of introduction: "one of the leading cause" should be substituted with "one of the leading causes".

- Table 2: there is a typo in the title. It should be "studies" not "stud".

- Figures 1 and 2: The phrase "The original figure was made for this review using the Canva software by Radka Michalkova." should be deleted.

Reviewer 2 Report

in this review, the authors describe the potential anti-cancer effect of natural chalcones.

The bibliography has been the subject of considerable work, but it is far too long (72 pages and 373 bibliographical references) and repetitive.

Authors really need to condense and not write a paragraph for each article referenced in this review and select the most relevant articles.

This review needs to be synthesized and rewritten according to the summary tables for each chalcone (with sections by compound and not by pathology), because this is repetitive, for example the effects of molecules such as Xanthohumol, which inhibit cell proliferation in all types of cancer cited.

I cannot accept this review in this writing format and with this number of pages.

Reviewer 3 Report

The review paper is very well written. The authors clearly knew what and how they wanted to present. In the following chapters, they present the current view of the action of chalcones against breast cancer, digestive system cancer (pancreatic cancer, oral cancers, esophageal cancer and others), lung cancer, bladder cancer, renal cancer, prostate cancer, and melanoma. The authors focused on the review of the anti-cancer properties of compounds such as: xanthohumol, butein, isoliquiritigenin, cardamonin, garcinol, flavokawains, isobavachalones, licochalcones, flavokawain B, panduratin A, broussochalcone A. The authors do not provide only IC50 values but focus directly on the mechanisms of action of the above-mentioned compounds. In addition, the results of in vivo/in vitro studies are summarized in tables. One note about the tables - their structure/format should be uniform throughout the work. Particularly noteworthy is the fact that the extensive (373 positions) cited literature is the latest, mainly from the last 5 years. I think that from the substantive side, the work is written very well, although I do not think that garcinol belongs to chalcones. It is more correct to write that garcinol is an analog or derivative of chalcones. From the editorial point of view, the above-mentioned tables require improvement. On line 449 "trans" should be written in italic. In addition, I would like to ask you to place the structures of all the discussed chalcones.

English language without any major complaints. Written correctly in the style in which scientific articles should be written. Small stylistic mistakes.

Round 2

Reviewer 2 Report

The authors have made the required changes.

I accept in this present form.